behaviour, cognition, developmental biology

language acquisition, hierarchy, power law, language evolution

**Author for correspondence:**
Tim Sainburg
e-mail: timsainb@gmail.com

# Long-range sequential dependencies precede complex syntactic production in language acquisition

Tim Sainburg[1,2], Anna Mai[3] and Timothy Q. Gentner[1,4]

[1]Department of Psychology, [2]Center for Academic Research & Training in Anthropogeny, [3]Department of Linguistics, and [4]Neurosciences Graduate Program, Neurobiology Section, Kavli Institute for Brain and Mind, UC San Diego, La Jolla, CA 92093, USA

 TS, 0000-0003-4223-2689; AM, 0000-0002-8343-9216; TQG, 0000-0002-4516-9841

To convey meaning, human language relies on hierarchically organized, long-range relationships spanning words, phrases, sentences and discourse. As the distances between elements (e.g. phonemes, characters, words) in human language sequences increase, the strength of the long-range relationships between those elements decays following a power law. This power-law relationship has been attributed variously to long-range sequential organization present in human language syntax, semantics and discourse structure. However, non-linguistic behaviours in numerous phylogenetically distant species, ranging from humpback whale song to fruit fly motility, also demonstrate similar long-range statistical dependencies. Therefore, we hypothesized that long-range statistical dependencies in human speech may occur independently of linguistic structure. To test this hypothesis, we measured long-range dependencies in several speech corpora from children (aged 6 months–12 years). We find that adult-like power-law statistical dependencies are present in human vocalizations at the earliest detectable ages, prior to the production of complex linguistic structure. These linguistic structures cannot, therefore, be the sole cause of long-range statistical dependencies in language.

## 1. Introduction

Since Shannon's original work characterizing the sequential dependencies present in language, the structure underlying long-range information in language has been the subject of a great deal of interest in linguistics, statistical physics, cognitive science and psychology [1–20]. Long-range information content refers to the dependencies between discrete elements (e.g. units of spoken or written language) that persist over long sequential distances spanning words, phrases, sentences and discourse. For example, in Shannon's original work, participants were given a series of letters from an English text and were asked to predict the letter that would occur next. Using the responses of these participants, Shannon derived an upper bound on the information added by including each preceding letter in the sequence. More recent investigations compute statistical dependencies directly from language corpora using either correlation functions [3,4,7,8,10,12,13] or mutual information (MI) functions [2,5,6,14] between elements in a sequence. In both cases, dependencies are calculated as a function of the sequential distance between pairs of elements. For example, in the sequence $a \rightarrow b \rightarrow c \rightarrow d \rightarrow e \rightarrow f$, at a distance of three elements, relationships would be calculated over the pairs $a$ and $d$, $b$ and $e$ and $c$ and $f$.

On average, as the distance between elements increases, statistical dependencies weaken. Across many different sequence types, including phonemes, syllables and words in both text and speech, the decay of long-range correlations and MI in language follows a power law (equation (2.6)) [2–14,18,19]. This power-law relationship is thought to derive at least in part from the hierarchical organization of language, and has been variously attributed to

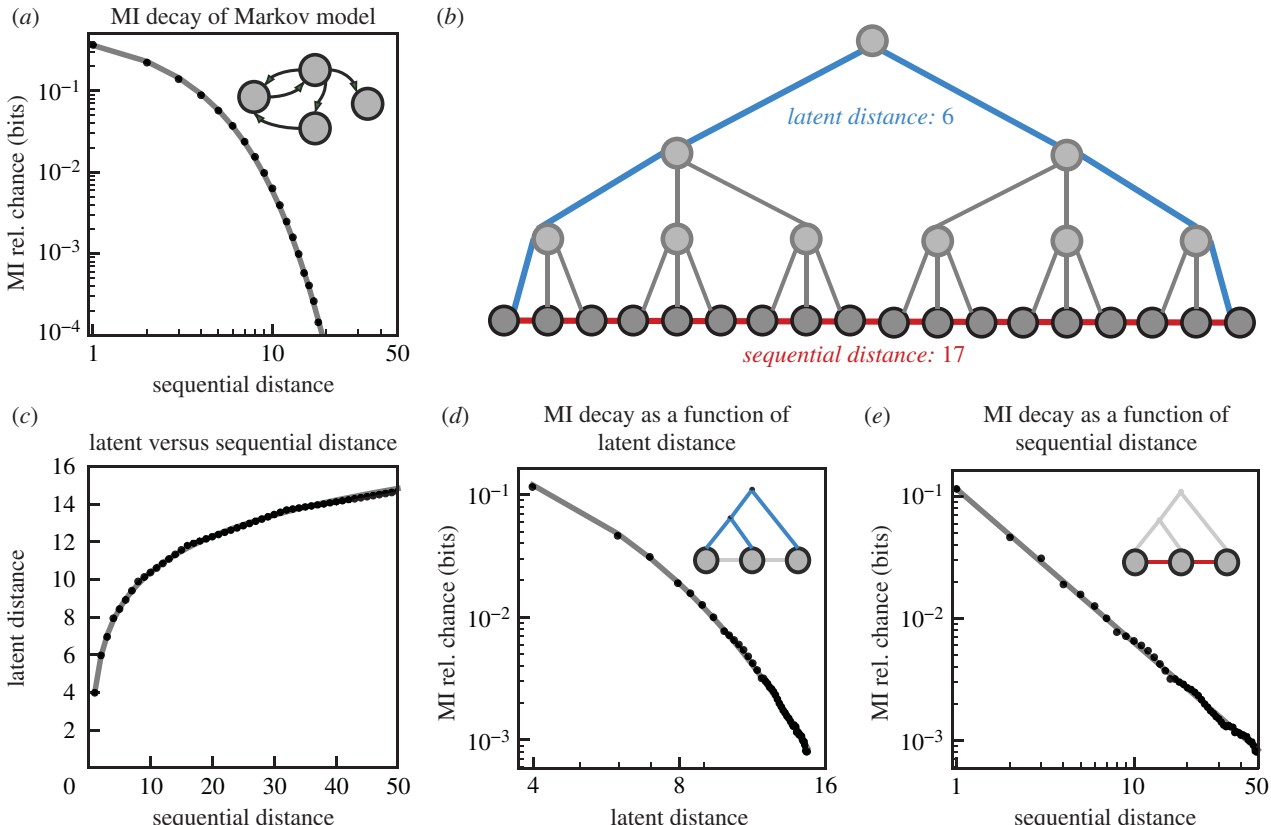

**Figure 1.** Comparison of long-range statistical dependencies between sequences with and without deep latent relationships. (a) The MI between elements in an iteratively (Markov model) generated sequence decays exponentially as a function of sequential distance. (b) An example sequence with hierarchical latent structure. The latent distance between the two end elements in the sequence is 6 (blue), while the sequential distance is 17 (red). (c) In sequences with hierarchical latent structure, the sequential distance between elements is logarithmically related to the latent distance (fit model: $a*\log_{x*b} + c$ where $x$ is sequential distance). (d) Like sequential distance in (a), the MI between elements in a hierarchically generated sequence decays exponentially as a function of latent distance. (e) The MI between elements in a hierarchically generated sequence decays following a power law as a function of sequential distance, which is related to the exponential MI decay seen in (d) and the logarithmic relationship between sequential and latent distance seen in (c). In (a), the probabilistic Markov model used to generate the empirical data has two states with a self-transition probability of 0.1. In (c–e), a probabilistic context-free grammar [5] with the same transition probability is used.

hierarchical structure in human language syntax [5], semantics [3] and discourse structure [4].

## (a) Mechanisms for long-range organization

To understand the link between hierarchical sequential organization in language and long-range sequential dependencies, it is helpful to consider both the latent and surface structure of a sequence (figure 1). When only the surface structure of a sequence is available, as it is for language corpora, a power-law decay in the MI between sequence elements gives evidence of an underlying hierarchical latent structure [5]. This phenomenon can be demonstrated by comparing the MI between elements in a sequence generated from a hierarchically structured language model, such as a probabilistic context-free grammar (PCFG), to the MI between elements in a sequence generated by a non-hierarchical model, such as a Markov process (figure 1). For sequences generated by a Markov process, the strength of the relationship between elements decays exponentially (equation (2.5)) as sequential distance increases [5,21] (figure 1a). In the PCFG model, however, linear distances in the sequence are coupled to logarithmic distances in the latent structure of the hierarchy (figure 1b,c). While information continues to decay exponentially as a function of the distance in the latent hierarchy (figure 1d), this log-scaling results in a power-law decay when MI is computed over corresponding sequential distances (figure 1e).

## (b) Hierarchy in language and behaviour

The thought that human language syntax is generated by CFGs [22] has led many to speculate that the long-range dependencies observed in language corpora are the product of abstract linguistic structure [2–5]. Although the long-range statistical dependencies in language corpora are clearly tied to linguistic structure [3,4], it does not follow that syntactic structure is necessarily the only source for long-range dependencies in language. Indeed, hierarchical organization is unique to neither CFGs nor human language and diverse classes of mechanisms, many of which are decidedly not language-like [23–28], are capable of generating power-law relationships. Many non-linguistic human behaviours [29–34], animal behaviours [35–40], animal vocalizations [41–50] and other biologically generated processes [28,51–57] are organized hierarchically. Likewise, long-range, power-law distributed dependencies are observed in sequential behaviours, including whale song [46], birdsong [2,58] and *Drosophila* [39] and zebrafish motility [59]. Instead, long-range dependencies in language and other human behaviour [33,60,61] may reflect more general biological processes inherited from the organization of underlying neurophysiological mechanisms [62–65]

that are, in turn, characterized by power-law relationships in temporal sequencing [66–68]. When viewed as an instance of this more general class of sequentially organized behaviour, one might reasonably predict that human speech should display long-range statistical dependencies independent of linguistic structure.

### (c) General hierarchical organization, linguistic hierarchy and hierarchical description

While many biological, behavioural and other natural signals can be and have been described using hierarchical terms, an important distinction exists between the many different ways in which the term hierarchy is used. For example, the generative mechanisms underlying some signals are either known or widely thought to be organized hierarchically, such as the cascade of motor programs originating in cortex and ending in affector neurons or the production rules underlying formal language theory. In other signals, hierarchicality is less clear, and what is often used to determine whether a signal is hierarchical is whether that signal is well described using hierarchical terms, without explicit hypotheses about the source of that hierarchy. For example, compression via tree-based grammars like the one in figure 1 are commonly used to describe hierarchical organization of non-linguistic structures such as animal behaviours and genomic and protein sequences [59,69,70].

In this work, we specifically explore the relationship between long-range statistical dependencies in speech and the emergence of hierarchical linguistic organization (i.e. syntactic, semantic and discourse structure). We ask whether the long-range statistical dependencies present in speech originate alongside the presence of syntactically complex linguistic productions, or whether they precede the production of this specific form of hierarchical structure during development. The developmental emergence of hierarchical organization in linguistic productions is well studied [71–73]. Meylan *et al.* [72], for example, demonstrate that linguistic productivity is initially low and rapidly develops at around 24 months of age, suggesting that young children's speech lacks the rich grammatical structure that enables language's remarkable productivity when they are first producing language. Hierarchical organization in semantic and discourse structure also grow in hierarchical complexity throughout ontogeny [71]. By 5–6 years of age, children are capable of producing well-formed narratives, which continue to increase in complexity to adulthood [74,75]. Thus, if long-range statistical dependencies are observed at an age prior to the emergence of complex linguistic structure, at both syntactic and discourse levels, those dependencies are not driven by those levels of linguistic structure and are therefore not likely to be driven fully by those linguistic structures in adult speech either.

### (d) Origins of long-range statistical dependencies in language

To test whether long-range statistical dependencies occur independently of complex linguistic structure in speech, we used MI decay as a measure of long-range dependencies over several speech corpora from children ranging from six months of age to adults [76–89]. Because complex linguistic productions emerge during language acquisition, we use these corpora to determine whether long-range relationships are present in human vocalizations prior to the production of linguistically complex speech, or whether they emerge alongside linguistically complex productions. If long-range dependencies were to emerge over the course of development alongside complex utterances, we could conclude that abstract linguistic structure plays a dominant role in the sequential statistical structure of speech. However, if long-range statistical dependencies are observed in infant speech prior to the production of structurally complex utterances, then it is likely that the long-range dependencies observed in adult speech are not solely governed by abstract linguistic structure. Indeed, we find that human speech exhibits long-range power-law statistical dependencies like those observed in mature human language early in development, at 6–12 months of age, while children are still in the 'babbling' stage of language development.

## 2. Methods

### (a) Datasets

We examined MI decay in sequences of words over nine datasets of natural speech from English speaking children included in the CHILDES repository [77,82–89] and three datasets of sequences of phonemes from the PhonBank repository [76,78–80], both of which are part of the TalkBank repository [77]. Each dataset within CHILDES and PhonBank was collected in a slightly different manner. In our analyses, we included only transcripts of spontaneous speech that were collected from typically developing children (usually at an in-home setting with family or an experimenter). The subset of CHILDES we used includes word-level transcripts of speech from children aged 12 months to 12 years of age. The subset of PhonBank we used includes phonetic transcriptions of speech given in the International Phonetic Alphabet (IPA) from children aged six months to four years of age. Between the phoneme and word-level datasets, a large range of speech and language development is covered.

For the MI analysis on phonemes, we binned transcripts into five 6-month age groups (6–12, 12–18, 18–24, 24–30, 30–36) and one age group from 3 years to 4 years. Each transcript was analysed as sequences of phonemes, where phoneme distributions for each transcript are treated independently to account for variation in acquired vocabulary across individuals during development. Because transcript lengths varied between age groups (electronic supplementary material, figure S1), we analysed MI at sequential distances up to the median transcript length for each age group. For the MI analysis on words, we binned transcripts into four 6-month age groups (12–18, 18–24, 24–30, 30–36) and one age group from 3 years to 12 years. We analysed words in the same manner as phonemes. No 6–12 month age group was used in word-level analyses because of the sparsity of word-level productions at that age. We additionally repeated our analysis on a set of French transcripts gathered from PhonBank [90–95].

### (b) Transition entropy and Sequitur hierarchical compressability

Before performing the MI decay analysis we looked at the transition structure and hierarchical compressibility for the 25 longest PhonBank transcripts for each age group to determine whether any sequential structure is present in speech at each age group and if so to quantify and compare how well each speech dataset can be described hierarchically.

To quantify whether non-random transition structure is present in the vocalizations of each age group, we computed a transition matrix between phonemes in each corpus (i.e. a first-order Markov model). We then compared the non-randomness

of the sequence on the basis of the mean transition entropy across states. This transition entropy captures the level of uncertainty about the next state, given the current state. For example, an element (A) that transitions to another element (B) 100% of the time will have a transition entropy of 0. Thus, if the mean transition entropy of a transcript is lower than the mean transition entropy of a randomized transcript, there is some degree of predictability in that transcript beyond randomness. As speech at all ages is dictated both by motor constraints and experience with language starting *in utero* [96], we expect non-random structure to be observed at all ages.

We then quantify how hierarchically compressible each transcript is relative to a transcript of equal length generated by a Markov model. To quantify the hierarchical compressibility, we use the Sequitur algorithm [69] which infers a hierarchical re-write system from symbolic sequences. Sequitor infers a form of deterministic grammar that is a restriction on context-free grammars and does not allow recursion or the capacity for a symbol to take more than one expansion from a non-terminal [97]. Sequitur has been used to losslessly and compressively represent data spanning linguistic, biological, behaviour and generally hierarchical signals (see [69] for the algorithm) and can be used to compare the relative compressibility of sequences [70]. We compare the hierarchical compressibility of transcripts to Markov-generated sequences as controls for each transcript.

## (c) Mutual information analysis over sequences

For each dataset, we calculate the sequential MI over the elements of the sequence dataset (e.g. words produced by a child). Each element in each sequence is treated as unique to that transcript to account for different distributions of behaviours across different transcripts within datasets.

Given a sequence of discrete elements $a \to b \to c \to d \to e$. We calculate MI as:

$$I(X, Y) = S(X) + S(Y) - S(X, Y), \tag{2.1}$$

where $X$ and $Y$ are the distributions of single elements at a given distance. For example, at a distance of two, $X$ is the distribution $[a, b, c]$ and $Y$ is $[c, d, e]$ from the set of element-pairs $(a - c, b - d$ and $c - e)$. $\hat{S}(X)$ and $\hat{S}(Y)$ are the marginal entropies of the distributions of $X$ and $Y$, respectively, and $\hat{S}(X, Y)$ is the entropy of the joint distribution of $X$ and $Y$.

To estimate entropy, we employ the Grassberger [98] method which accounts for under-sampling true entropy from finite samples:

$$\hat{S} = \log_2(N) - \frac{1}{N} \sum_{i=1}^{K} N_i \psi(N_i), \tag{2.2}$$

where $\psi$ is the digamma function, $K$ is the number of categories of elements (e.g. words or phones) and $N$ is the total number of elements in each distribution.

We then adjust the estimated MI to account for chance. To do so, we subtract a lower bound estimate of chance MI ($\hat{I}_{sh}$):

$$MI = \hat{I} - \hat{I}_{sh} \tag{2.3}$$

This sets chance MI at zero. We estimate MI at chance ($\hat{I}_{sh}$) by calculating MI on permuted distributions of labels $X$ and $Y$:

$$\hat{I}_{sh}(X, Y) = \hat{S}(X_{sh}) + \hat{S}(Y_{sh}) - \hat{S}(X_{sh}, Y_{sh}). \tag{2.4}$$

$X_{sh}$ and $Y_{sh}$ refer to random permutations of the distributions $X$ and $Y$ described above. Permuting $X$ and $Y$ effects the joint entropy $S(X_{sh}, Y_{sh})$ in $I_{sh}$, but not the marginal entropies $S(X_{sh})$ and $S(Y_{sh})$.[1] $\hat{I}_{sh}$ is related to the expected mutual information [99–101] which accounts for chance using a hypergeometric model of randomness.

Importantly, MI calculated over a sequence as a function of distance is referred to as a 'MI function', to distinguish it as the functional form of MI, which measures the dependency between two random variables [14]. In the MI function, samples from the distributions $X$ and $Y$ are taken from the same sequence, thus they are not independent. MI as a function of distance acts as a generalized form of the correlation function that can be computed over symbolic sequences and captures nonlinear relationships [14].

## (d) Fitting mutual information decay

We fit the three following models: An exponential decay model:

$$MI = a * e^{-x*b} + f. \tag{2.5}$$

A power-law model:

$$MI = c * x^d + f. \tag{2.6}$$

A composite model of the power-law and exponential models:

$$MI = a * e^{-x*b} + c * x^d + f, \tag{2.7}$$

where $x$ represents the inter-element distance between units (e.g. phones or syllables).

To fit the model on a logarithmic scale, we computed the residuals between the log of the MI and the log of the model's estimation of the MI. We scaled the residuals during fitting by the log of the distance between elements to emphasize fitting the decay in log-scale because distance was necessarily sampled linearly as integers. Models were fit using the lmfit Python package [102] using Nelder–Mead minimization. We compared model fits on the basis of AICc and report the relative probability of each model fit to the MI decay [2,103].

## (e) Mutual information decay controls

Datasets are organized hierarchically into transcripts, utterances, words and phonemes allowing us to shuffle the dataset at multiple levels of organization. To ensure that our MI decay results are a direct result of the sequential organization of each dataset, we performed a control on each dataset shuffling behavioural elements within each individual transcript at each hierarchical level. In addition, to ensure that long-range relationships were not due to trivial repetitions of behaviours, for each dataset we looked at MI decay over sequences in which repeated elements were removed. Finally, we analysed transcripts from a subset of the longest individual transcripts to confirm that our results were not the product of mixing together multiple datasets and transcripts.

# 3. Results

## (a) Transition entropy and hierarchical compression

We analysed the transition entropy and hierarchical compressibility of the 25 longest transcripts for each age group in the PhonBank datasets. An example Markov model inferred from three of these age groups is given in electronic supplementary material, figure S2. Across each age group, as expected, we found non-random transition structure indicated by the mean transition entropy (figure 2a).

We then compared the Sequitur hierarchical compressibility of each transcript to a length-controlled Markov-generated transcript as well as a randomly permuted version of the original transcript. Because in our datasets transcripts vary in size, with younger age groups typically having shorter transcripts available, we artificially varied the length of

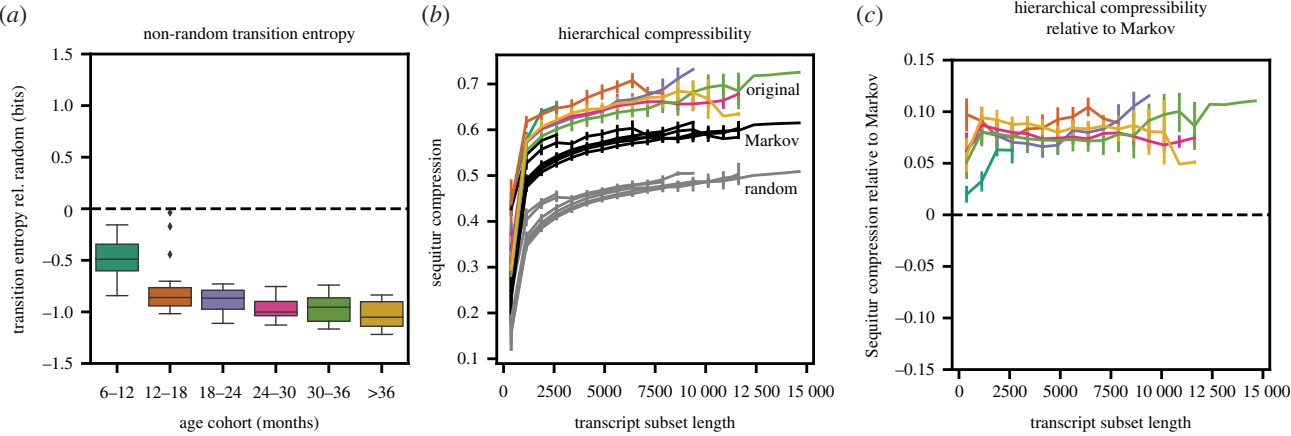

**Figure 2.** Transition entropy and hierarchical compressability of children's speech. (*a*) The average transition entropy of the 25 largest phoneme corpora for each age group, relative to the transition entropy of a permutation of that corpora. The dashed line indicates equal transition entropy. Values below the dashed line indicate transitions are more predictable than random sequences. (*b*) Sequitur hierarchical compressibility (compressed sequence length divided by original sequence length) for the same transcripts in (*a*), the permutations in (*a*), and matched-lengthed transcripted generated from Markov models trained on the transcripts in (*a*). (*c*) The same Sequitur compression results as in (*b*) plotted relative to the Markov results in (*b*). Values above the dashed line indicate greater hierarchical compressability than a first-order Markov model. (Online version in colour.)

each transcript between 1 and 100% of the full transcript and computed the compressibility of the transcript at each length to enable comparison between age-groups.

We find that, across all age groups, transcripts are more hierarchically compressible than their Markov or randomized counterparts (figure 2*b*). We also find that the degree to which these transcripts are compressible over their Markov counterparts is similar across each age group (figure 2*c*). These results indicate non-random and hierarchical structure across corpora from all age groups. In the following MI analyses, we measure the sequential relationships underlying these signals.

## (b) Mutual information

Although much work has explored the information content and long-range sequential organization of human language, relatively few studies have examined these properties in speech [2] or language development directly. Here we investigate the long-range information present in speech during language development using datasets from the TalkBank project [76,77].

We examined MI decay in sequences of phones over three datasets of natural speech from English-speaking children included in the PhonBank repository. Across all age groups, starting at 6–12 months of age, the decay in MI over sequences of phonemes is best fit by a composite power-law and exponential decay model (figure 3*a*–*c*; relative probabilities 0.897 to > 0.999; electronic supplementary material, table S1). In each age group, we observe both a clear power law, prominent over long distances (figure 3*b*), and a clear exponential decay at short, word-length distances (figure 3*c*), consistent with prior results adult speech [2]. We then examined MI decay in sequences of words over nine datasets of natural speech from English-speaking children included in the CHILDES repository. As with phonemes, the MI decay between words is best fit by a composite model of power-law and exponential decay (equation (2.7); relative probability = 0.989 for 12–18 months and greater than 0.999 for all other age groups; figure 3*d*–*f*; electronic supplementary material, table S2). To ensure that our results are not specific to English, we repeated this analysis over French corpora also taken from PhonBank and

found the same power-law and exponential components of the decay in each age group (electronic supplementary material, figure S3). The parameters for each best-fit model for figure 3 can be found in electronic supplementary material, table S3. We additionally plotted the decay parameters of the PhonBank datasets for individual corpora across age groups varying corpus size in electronic supplementary material, figure S4.

## (c) Mutual information decay controls

As controls, we also computed the MI decay over sequences of words and phonemes that had been shuffled to isolate sequential relationships at different levels of organization (e.g. phoneme, word, utterance, transcript). A subset of these controls over the PhonBank dataset are shown in figure 4 while the remainder are given in electronic supplementary material, figures S5 and S6.

To aid in interpreting these controls, we additionally performed equivalent shuffling controls with the PCFG model from figure 1 extended with Markov-generated endpoints (figure 4*a*; as in Sainburg *et al.* [2]). In this model, we replace each terminal state generated from the PCFG with a Markov-generated sequence. These two separate models (hierarchical and finite-state) capture the distinction between the finite-state, Markovian, organization of phonological relationships at short distances (e.g. within and between words) [104] and the hierarchical organization which extends beyond (e.g. syntax, discourse). We have previously found that this model captures the observed interplay between the exponential and power-law MI decay found in speech [2], where the exponential decay is driven by the Markovian model and the power-law is driven by the context-free grammar.

Broadly, across each age group, we find results that are consistent with the model from figure 4*a*–*e*. We observe that short-range relationships captured by exponential decay are largely carried within words and utterances, while long-range relationships are captured by a power-law decay carried across longer timescales between words and utterances. In the full model, as in figure 3*a*, the MI decay (figure 4*b*) is well explained by an exponential decay occurring at shorter

*Proc. R. Soc. B* **289**: 20211657

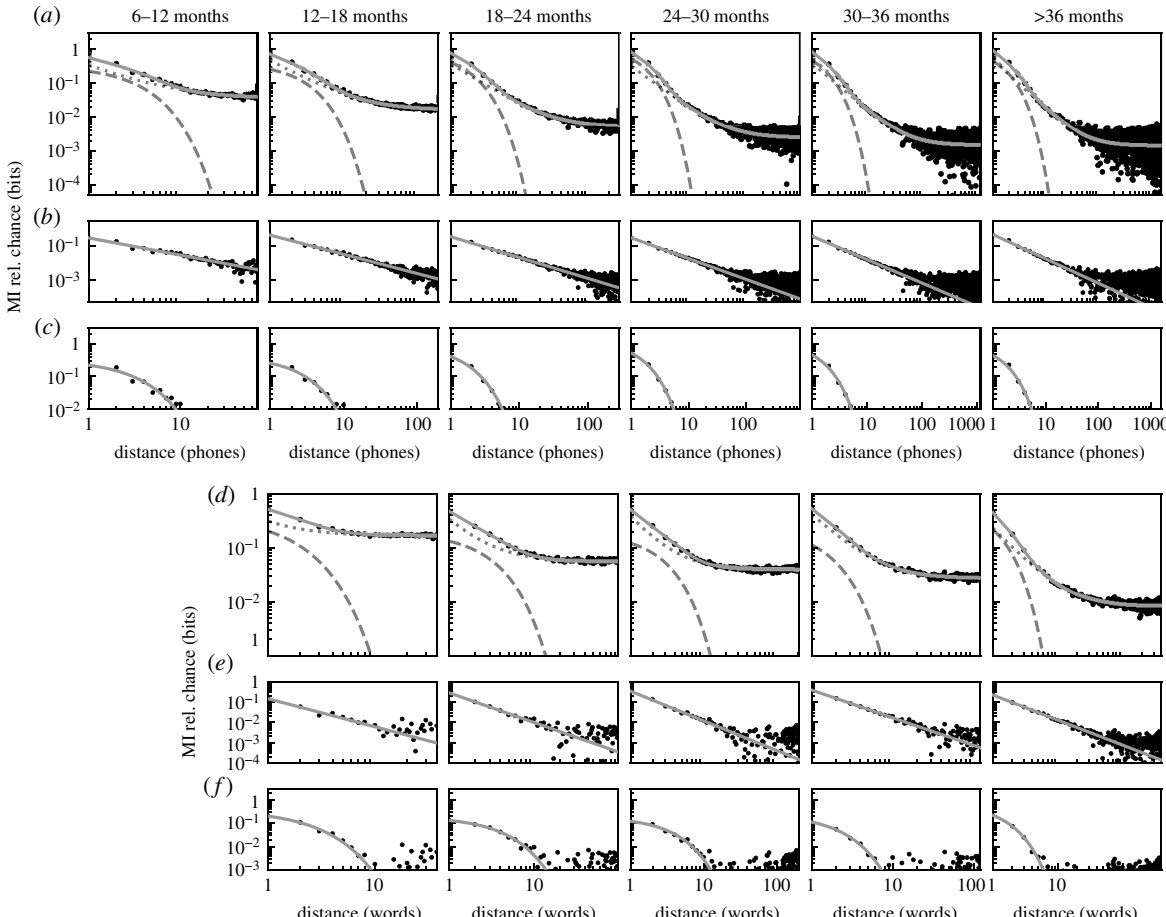

**Figure 3.** MI decay over words and phonemes during development. (*a*) MI decay over phonemes for each age group. MI decay is best fit by a composite model (solid grey line) for all age groups across phonemes and words. Exponential and power-law decays are shown as a dashed and dotted grey lines, respectively. (*b*) The MI decay (as in (*a*)) with the exponential component of the fit model subtracted to show the power-law component of the decay. (*c*) The MI decay (as in (*a*)) with the power-law component subtracted to show the exponential component of the decay. (*d–f*) The same analyses as (*a–c*), but for words. No 6–12 month age group was used in word-level analyses because of the sparsity of word-level productions at that age.

distances (dashed line) and a power-law decay at longer distances (dotted line). Shuffling at different levels of organization modifies the relative contribution of these exponential and power-law components of the decay.

Shuffling Markov sequences within the transcript (figure 4*c*) relates to shuffling words within the transcript (figure 4*g*). Because the hierarchical structure is eliminated but the Markovian structure is preserved, the power-law component of the decay is effectively eliminated (figure 4*c*, dashed line). Likewise, because within-word structure is largely preserved but between word structure is destroyed, the power-law component of the decay is also eliminated (figure 4*g*).

On the opposite end of the hierarchy, when Markovian structure is destroyed by shuffling the sequence within the Markov-generated sequences (figure 4*d*), the exponential component of the decay is lost (dashed line) and the decay is dominated by the power law. Similarly, when phones are shuffled within utterances (figure 4*h*), the power-law component of the decay is largely retained while the exponential component is reduced (relative to figure 3*a*). Consistent with the analyses over words in the CHILDES transcripts (figure 3*f*), the exponential component of the decay is not entirely destroyed when shuffling within utterances (figure 4*h*) or words (electronic supplementary material, figure S5*c*), indicating that a boundary between Markovian and hierarchical organization cannot be fully assigned at either level of organization.

Finally, in an intermediary shuffling, as in the mid-level branch sequences (figure 4*e*), the model is still well explained by both the exponential and power-law decay components, however, the power-law component (dotted line) now falls between what it did when all of the hierarchical structure was destroyed (figure 4*c*) and when the full signal was present (figure 4*b*). Again, we observe similar results with the transcript data in figure 4*i* (in comparison to figure 3*a* and figure 4*g*). A complete set of shuffling results at more levels of organization as well as with the CHILDES dataset are given in electronic supplementary material, figures S5 and S6.

As an additional control, to ensure that the observed MI decay patterns are not the product of mixing datasets from multiple individuals, we also computed the MI decay of the longest individual transcripts comprising each age cohort across both phonemes and words. The decay of the longest individual transcripts parallel the results across transcripts shown in figure 3 (electronic supplementary material, figures S7 and S8). We also analysed the MI decay of transcripts when repeated elements were removed to ensure long-range relationships were not the product of behavioural repetitions (as in e.g. [41]). Removing repeats does not qualitatively alter the pattern of long-range relationships between elements (electronic supplementary material, figure S9).

One reasonable hypothesis is that these long-range relationships in child speech are driven by interaction. Child speech in

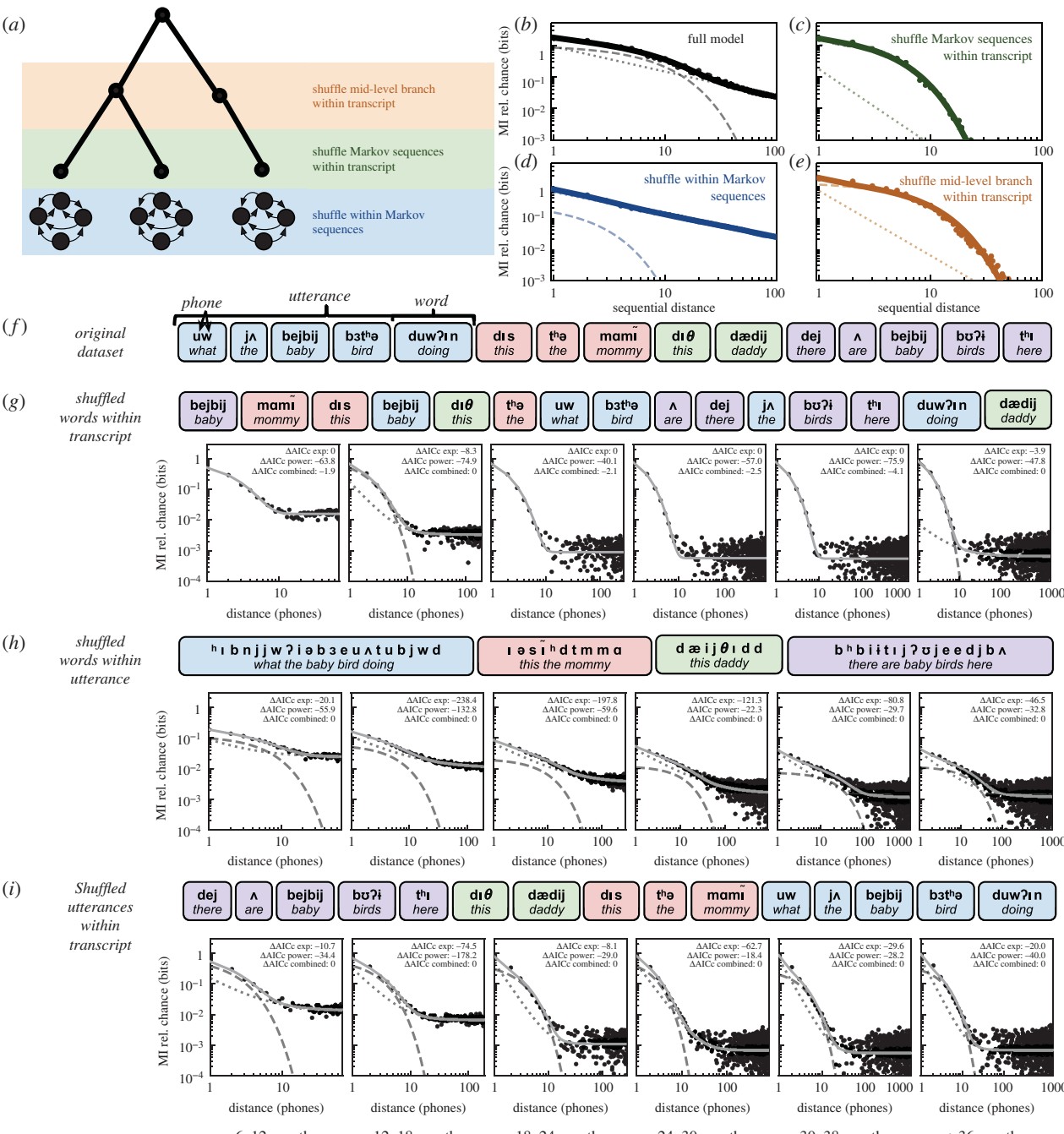

**Figure 4.** MI decay between phones under different shuffling conditions. (*a*) An example PFSG model with endpoints generated from Markov models as in [2]. Shuffling controls are given at three levels of organization in the model: within Markov-generated sequences (bottom/blue), between Markov-generated sequences (middle/green) and at a mid-level branch in the generated hierarchy (top/orange). (*b*) The MI decay curve for the model from (*a*). (*c*) The MI decay curve for the shuffle Markov sequences condition from (*a*). (*d*) The MI decay curve for the shuffle within Markov sequences condition from (*a*). (*e*) The MI decay curve for the shuffle within mid-level branch condition from (*a*). (*f*) An example sequence of utterances from the PhonBank dataset. Utterances are grouped by colour, words are grouped by rounded rectangles, and phones are displayed in bold above orthographic transcriptions. (*g*) MI decay, as in figure 3 when words are shuffled within each transcript. ΔAICc values (relative to best fit model) are shown for each fit model in each panel. (*h*) MI decay when phones are shuffled within each utterance. (*i*) MI decay when utterances are shuffled within each transcript. For each panel, as in figure 3*a,d*, the dashed and dotted lines represent the exponential and power-law components of the fit composite decay, respectively. When the best-fit model is not the combined model, a solid line is shown for the model fit without the dashed or dotted components.

the datasets used here is produced in an interactive context with adults, which affects discourse structure in children's speech [105] thus, adult speech could be driving the long-range relationships observed in child speech. If this were the case, one could argue that the complex hierarchical structure underpinning the adult's speech was driving the long-range dependencies found in infant speech. To test whether this is the case, for each corpus where adult speech was transcribed ($n_{\text{CHILDES}} = 1630$, $n_{\text{PhonBank}} = 309$) we tested the effect of

non-subject engagement (the proportion of speech not produced by the child) on the improvement in model fit (ΔAICc) of a power-law model over exponential model alone. In both datasets, we observe that adult involvement provides no additional predictive information about the improvement in fit of the power-law model over the exponential model, when controlling for the dataset, child's age and length of the transcript (CHILDES: $F_{1,1620} = 1.49$, $p = 0.22$; PhonBank: $F_{1,306} = 0.21$, $p = 0.65$). Although our results do not provide irrefutable

evidence that the long-range range relationships observed are driven by adult speech, these results do not rule out the possibility. Our analyses were based on the natural variability in adult speech across corpora and are not explicitly controlled.

## 4. Discussion

We analysed the developmental time course of long-range sequential information present in speech throughout early childhood. We observed adult-like long-range statistical relationships [2] present as early as 6–12 months in phoneme sequences, and at 12–18 months in word sequences. Our findings compel reconsideration of the mechanisms that shape long-range statistical relationships in human language. Adult speech is characterized by both long-range and short-range statistical relationships characterized by power-law and exponential decays in MI, respectively. Traditionally, the power-law decay in information between the elements of language (phonemes, words, etc.) has been thought to be imposed by the hierarchical linguistic structure of syntax, semantics, and discourse [3–5]. Early speech development provides a natural experiment in which one can examine human vocal communication absent the production of complex syntactic and discourse structure [71–75]. Remarkably, even at a very early age, prior to the production of mature linguistic structures, vocal sequences show adult-like long-range dependencies.

The long-range dependencies observed in adult language corpora have been clearly tied to linguistic structure [3–5], but the absence of these organizational components in productions of the youngest children rule out the possibility that linguistic structure is the only driver of long-range dependencies in human speech. One reasonable hypothesis consistent with our results is that multiple mechanisms impose long-range dependencies on human speech and language and that these operate on different developmental timescales. Below we consider non-linguistic mechanisms that could generate long-range dependencies in speech at early ages. Limitations of the current dataset prevent quantitative exploration of these mechanisms, but we note how they could be pursued in future research.

One possibility is that the long-range structure we observe in young children is driven by proximal environmental factors, such as the long-range statistics in a child's linguistic environment. In animal behaviour, long-range statistical relationships between behavioural states can be affected by seemingly inconsequential variables such as lighting environment [59]. In the child's environment, linguistic behaviour could be influenced by similar environmental variables. Furthermore, communication is an inherently interactive behaviour with other humans. While our results did not provide evidence that long-range relationships are driven by interactions with adult speakers, we cannot rule out language interaction or other exogenous variables as possible drivers for the observed long-range relationships. It would be valuable to explore the impact of environment on the parameters of the observed information decay (e.g. the exponential and power-lay decay parameters and transition between exponential and power-law decay). These factors cannot be properly adjudicated in the present study because our study relies on datasets where corpus variability across environments and age groups makes it difficult to compare model fits directly (e.g. electronic supplementary material, figure S4).

Another possible non-linguistic source of the observed long-range relationships in early speech is the general hierarchical organization of the motor control systems that produce speech and all other overt behaviours. Consistent with this, we note the observation of similar long-timescale dynamics in recent computational ethology studies examining comparatively simple behaviours such as *Drosophila* motility [39]. From this perspective, our results are consistent with the notion that specialized, hierarchical linguistic structure is overlaid on a more general (non-linguistic) hierarchical motor control structure. Accordingly, long-range dependencies relevant to language may emerge more slowly with the development of adult-like linguistic competence. Little is known at present, however, about the neural and motor control underpinnings of these long-range statistical dependencies in non-vocal behavioural sequences, and/or whether these relationships are driven by environmental structure.

The observation of similar power laws in diverse non-linguistic behaviours reinforces the idea that multiple mechanisms can shape the sequential dynamics of behaviour, including speech. There are many potential sources for long-range correlations in biological and physical systems that do not necessarily guarantee an underlying hierarchical structure [23–28]. For example, $1/f$ long-range statistical relationships are characteristic of both physiological and environmental signals. Although $1/f$ signals are often associated with hierarchical organization, their origins in physical systems remains an area of active research [27,106–108]. The presence of long-range correlations characterized as $1/f$ 'noise' in human neural signals is associated with healthy brain states and disappears in disease states and age-related impairments in working memory [108–110]. $1/f$ power spectra produce similar power-law correlations and MI decay as those we observe in speech; thus, it is possible that long-range statistical relationships in speech may originate from such physiological sources as those observed to generate $1/f$ power spectra. In electronic supplementary material, figure S10 we show that behavioural states tied to $1/f$ noise would produce the same power-law MI decay as the context-free grammar model from figure 1b,e. Comparing measurements of long-range statistical relationships in speech in both healthy and disease states in relation to the degradation of $1/f$ characteristics in neural signals would indicate whether long-range dependencies in speech and $1/f$ noise in neural systems are related.

Regardless of any further understanding of the specific mechanisms that underlie the sequential dependencies in speech, clear patterns in the information conveyed across time exist in human vocal behaviour at very early ages. In principle, this structure is available to listeners and can provide predictive information to any cognitive agent that engages with it. Humans are necessarily sensitive to long-range relationships in language, and although more sparse, evidence for long-range sensitivities outside language has also been reported, such as scale-invariance in retrospective memory tasks [111] and attention to power-law timescales in anticipation of future events in cognitive tasks [112]. Among non-human animals, the evidence supporting sensitivity to the long-range dynamics (power-law or otherwise) of information in the environment is not well studied, especially at long intervals. If non-human animals can perceive the long-range statistical dependencies present in their environment, this capacity may constitute a broad faculty of language

[113], that is, a necessary, but not uniquely human, component of language. Indeed, the presence of long-range statistical dependencies in non-linguistic behaviours and a generalized perceptual sensitivity to them could provide a scaffold for language to evolve, and where hierarchical syntax and semantics can be understood as later additions that exploit existing long-range structures and sensitivities.

Data accessibility. All scripts used in this study are openly accessible at https://github.com/timsainb/LongRangeSequentialOrgPaper. The data are provided in the electronic supplementary material [114]. The datasets can be acquired from the TalkBank repository [77] and PhonBank repository [76]. We performed analyses over these transcripts without any modification. Example transcripts for each dataset are displayed in the electronic supplementary material, Supplementary Information. The distribution of sequence lengths of each dataset is shown in electronic supplementary material, figure S1.

Authors' contributions. T.S.: conceptualization, data curation, formal analysis, funding acquisition, investigation, methodology, project administration, resources, software, validation, visualization, writing—original draft, writing—review and editing; A.M.: conceptualization, writing—original draft, writing—review and editing; T.Q.G.: conceptualization, funding acquisition, project administration, supervision, writing—original draft, writing—review and editing.

All authors gave final approval for publication and agreed to be held accountable for the work performed therein.

Competing interests. We declare we have no competing interests.

Funding. This work was supported by NSF GRF 2017216247 and an Annette Merle-Smith Fellowship to T.S., NIMH training fellowship T32MH020002 and William Orr Dingwall Dissertation Fellowship to A.M. and T.S., and NIH DC0164081 and DC018055 to T.Q.G.

## Endnote

[1]The joint entropy is computed over pairs of $X$ and $Y$, whereas the marginal entropy is computed over $X$ independently and $Y$ independently. Permuting $X$ and $Y$ shuffles the pairing between elements in $X$ and $Y$, but does not change the frequency of elements in the distributions $X$ or $Y$.

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
