## [Peer Review File · Proceedings of the Royal Society B: Biological Sciences]

Review History

RSPB-2021-0806.R0 (Original submission)

Review form: Reviewer 1

Recommendation

Major revision is needed (please make suggestions in comments)

Scientific importance: Is the manuscript an original and important contribution to its field?

Acceptable

General interest: Is the paper of sufficient general interest?

Good

Quality of the paper: Is the overall quality of the paper suitable?

Good

Is the length of the paper justified?

Yes

Should the paper be seen by a specialist statistical reviewer?

Yes

Do you have any concerns about statistical analyses in this paper? If so, please specify them explicitly in your report.

No

It is a condition of publication that authors make their supporting data, code and materials available - either as supplementary material or hosted in an external repository. Please rate, if applicable, the supporting data on the following criteria.

Is it accessible?

Yes

Is it clear?

Yes

Is it adequate?

Yes

Do you have any ethical concerns with this paper?

No

Comments to the Author

Summary and potential value of the contribution:

A fundamental property of human language is structural hierarchy. The evolution of hierarchical structure, and the degree to which there are analogs in non-human animals has been a subject of much research and debate. A challenge of such research, however, is that data from non-human animals is generally indirect (one cannot ask a humpback whale for grammaticality judgments). As such, statistical methods that track hierarchical structure may provide a useful way to directly compare humans to non-human animals.

The present paper investigates one such method, that looks at long-range dependencies between elements (using mutual information functions between elements of varying distances). For human language, long distance dependencies have been shown to follow a power law, which has been claimed to result from the hierarchical structure of human language. The contribution of the present paper is to throw this interpretation into question by showing that the same statistical pattern holds for productions of human infants that are assumed to be too young to have adult-like hierarchical structure.

General evaluation:

The general argument is interesting and potentially very useful to our understanding of the statistics of natural language. I nevertheless have some reservations with the paper in its present form. Of particular note, it currently lacks a positive proposal to explain the statistical patterns in question, which leaves the reader in a state of some mystery, not knowing how to interpret the results.

Main comments:

1. The paper would be significantly stronger if it could clearly and precisely outline a positive proposal -- a better model to explain the mechanisms by which such statistical patterns arise. If we are convinced by the arguments, then the patterns do not come from hierarchical organization of language. But then, what causes them? This being said, it is of course scientifically important to refute incorrect hypotheses. And the authors may decide that it is out of the scope of the present paper to provide a positive proposal to explain the facts. In this case, the degree to which the paper is publishable may depend on the degree to which the explanation in terms of hierarchy is widely assumed conventional wisdom. I have a hard time assessing this latter point.

2. I am not sure that what is being called hierarchy in this paper corresponds to the syntactic and phonological hierarchy that linguists have shown to exist for natural human language.

I am particularly confused by the results shown in Figure 3D. The authors show that shuffling phones within the utterance nevertheless allows long-range information (governed by the power law) to persist. The authors argue that this is problematic for the hierarchical interpretation of the pattern because it is showing up in children too young to have acquired complex linguistic structure. But the problem goes deeper than this. If similar results held for *adults*, it would also be an argument against interpreting this as linguistic hierarchy in any standard way. Specifically, the combinatorial systems of phonology and syntax are generally held to be two separate modules that largely do not interact (see, e.g. Pullum & Zwicky 1988; also relevant is the concept of "duality of patterning"). Thus, by scrambling the two systems in Figure 3D, both systems are completely disrupted. The only kind of "structure" remaining is discourse structure. But while there may be rhetorical relations that maintain discourse coherence, these are not generally taken to be the the same kind of linguistic structure that is arguably specific to humans (cf. Bolhuis et al. 2014).

Stated very coarsely, what the data in Figure 3D seem to show is that if one uses the sound /p/ in one utterance, one is more likely to use the sound /p/ in the following utterance. I don't believe that this follows from any principle of human phonology or syntax -- for infants or for adults.

3. Methodology: there is something paradoxical in the methods. The authors claim that children at 6-12 months are still in the babbling stage of development, so have no linguistic structure. But in Figure 3, the authors show that they shuffled words within the transcript, utterances within the transcript, and phones within the utterance -- even for children 6-12 months. But this presupposes that it is possible to identify words and utterances for children 6-12 months old. In other words, the authors identify three levels of hierarchy while at the same time asserting that hierarchy does not yet exist for this age group.

4. It could be useful to compare to other statistical methods that have been argued to provide evidence for syntactic combination. See, e.g. Yang (2013) for another way of characterizing the statistical profile of grammatical rules. Applying these methods to the sets of data analyzed here could provide stronger evidence for the (reasonable) assumption in the present paper that young children have not yet acquired complex linguistic structure. Ideally, what you would expect to find would be that the Yang measure becomes more adult-like over time, but that MI is adult-like from the beginning.

5. This paper should ideally be reviewed by someone who can check the statistical methods in more detail. I have just a couple of comments:

-Formula (1): shouldn't there be hats on the S's? S without a hat is never defined.

-Formula (2): for the distribution [a, b, c], I suppose that $N=3$ and $K=3$. N_i is not defined.

-It could be useful to provide simple examples of high and low entropy to give an intuition for how MI models long-distance dependency.

-I did not follow why "permuting X and Y [a]ffects the joint entropy ... but not the marginal entropies." If a simple example can be given briefly, this might be helpful.

Based on the shuffling manipulations, the paper makes the following generalization: "Across each shuffle analysis, we observe that short-range information content captured by exponential decay is largely captured within words and utterances, while long-range information is carried between utterances, even during early language production." While the graphs look somewhat different, I was unsure what the statistical basis for this conclusion was, given that the best model in many

cases was still the combined pow_exp model.

References

Bolhuis, Johan J., Ian Tattersall, Noam Chomsky, and Robert C. Berwick. 2014. How could language have evolved? PLoS Biology 12.e1001934.

Pullum, G. K., & Zwicky, A. M. (1988). The syntax-phonology interface. Linguistics: The Cambridge Survey: Volume 1, Linguistic Theory: Foundations, 1, 255.

Yang, Charles. (2013). Ontology and phylogeny of language. PNAS, 110(16): 6324-6327.

Review form: Reviewer 2

Recommendation

Major revision is needed (please make suggestions in comments)

Scientific importance: Is the manuscript an original and important contribution to its field?

Good

General interest: Is the paper of sufficient general interest?

Excellent

Quality of the paper: Is the overall quality of the paper suitable?

Excellent

Is the length of the paper justified?

Yes

Should the paper be seen by a specialist statistical reviewer?

No

Do you have any concerns about statistical analyses in this paper? If so, please specify them explicitly in your report.

No

It is a condition of publication that authors make their supporting data, code and materials available - either as supplementary material or hosted in an external repository. Please rate, if applicable, the supporting data on the following criteria.

Is it accessible?

Yes

Is it clear?

Yes

Is it adequate?

Yes

Do you have any ethical concerns with this paper?

No

Comments to the Author

General comments: Overall, this manuscript is clear, well-written, and interesting to read. The authors build off of their previous work in adult human speech and animal communication, and contextualize their research in the broader topic of sequential behaviors. The authors provide compelling evidence for long-range dependencies and hierarchical structure in the vocalizations of developing children that are reminiscent of those observed in adult speech.

Major comments:

While I think their findings are very interesting, I'd like the authors to speak more about the novelty of the work. Their results suggest that adult-like hierarchical structure can be observed even in the simple grammars of children: the utterances of children have some repeatable and predictable sequence structure, therefore can be described by Markovian and/or power-law decay functions. The authors are correct to highlight that hierarchical structure is also observed in non-linguistic human behaviors as well as in the vocalizations of non-human animals. So, how novel/unexpected is it that the speech patterns of children, even infants and toddlers, contain hierarchical structure?

The authors should discuss how the parameters of the power-law, exponential decay, and composite models compare between the current dataset and the corpus of adult speech from Sainburg et al. (2019), as well as age-dependent variation in the distance (in phones) at which the MI decay transitions from exponential to power law. I can appreciate the complexity of such comparisons across different levels of speech, but these comparisons would allow readers to assess how "adult-like" these long-range statistical relationships are in developing children.

The authors write that "long-range dependencies in language and other human behavior [33, 60, 61] may reflect more general biological processes inherited from the organization of underlying neurophysiological mechanisms [62–65]." In order to distinguish the contribution of linguistic input (learning) from fundamental ("innate") biological contributions, it would be revealing to compare the speech of at least one group of children that speak a different language. For example, because the shape of the MI functions seem to differ between adult English, German, and Japanese-speaking adults (Sainburg et al., 2019), comparing the English-speaking children to Japanese- or German-speaking children would be very informative. If the shape of the curve in children is a function of biology and not linguistic exposure, then the shapes should be statistically indistinguishable across children that speak different languages with different MI functions.

Minor comments:

In the last sentence of the abstract, the authors write that "(t)hese linguistic structures cannot, therefore, be the sole cause of long-range statistical dependencies in language." It is not really clear about the degree to which the authors are claiming that these long-range dependencies in infant vocalizations are innate. One question is to what degree long-range dependencies can be inferred from auditory patterns heard in utero. There is increasing evidence that auditory experiences in utero shape the brain and vocal output in humans, and it would be useful for the authors to discuss the extent to which in utero experiences could allow infants and toddlers to communicate with hierarchical structure.

Line 182-185: The authors should provide information about how prevalent repeated elements are at different levels (e.g., words, utterances) and different ages. This would provide some insight into the importance of this set of analyses.

Decision letter (RSPB-2021-0806.R0)

21-Jun-2021

Dear Mr Sainburg:

I am writing to inform you that your manuscript RSPB-2021-0806 entitled "Long-range sequential dependencies precede complex syntactic production in language acquisition" has, in its current form, been rejected for publication in Proceedings B.

This action has been taken on the advice of referees, who have recommended that substantial revisions are necessary. With this in mind we would be happy to consider a resubmission, provided the comments of the referees are fully addressed. However please note that this is not a provisional acceptance.

Sincerely,
Dr Robert Barton
<mailto:proceedingsb@royalsociety.org>

Associate Editor
Comments to Author:

This manuscript was reviewed by two experts and myself. We all found the manuscript to be interesting and potentially worthy of publication as it addresses important issues about language development. However, both reviewers raised serious concerns that prevent it from being published in its current form. It is not clear at this stage if the concerns can be adequately addressed. If they can be addressed, I would allow resubmission and attempt to have the manuscript reviewed by the same reviewers. Reviewer 1 points out a potential problem with the methods. How are there 3 levels of hierarchy in the speech of 6-12 month old children which are needed to perform the shuffling analyses shown in figure 3? A central claim is that these children lack these structures. Another problem raised by Reviewer 1 is that it is not clear what the shuffling of phones analysis (shown in 3D) is doing and how it might relate to the different

hierarchies present in adult speech. Repeating these analyses with adult speech or at least better integrating the work with other descriptions of adult speech (as suggested by Reviewer 2) and other methods of characterizing speech (as suggested by Reviewer 1) would help clarify the issue. This might allow you to better differentiate whether the long-range dependencies found here are stepping stones towards hierarchical speech (i.e., 'precede' adult speech) or are something unrelated. In addition, there is general confusion about how long-range dependency might relate to hierarchical structures. Is one necessary for the other? This relates to Reviewer 1's suggestion to include a positive explanation for what is happening--how can we get long-range dependency without hierarchical structure (e.g., is, as suggested by the example in figure 3, it just a matter of continuing to discuss the same topic which can lead to repeating phones over long periods of time?). Reviewer 2 also points out that looking at this in another language might help understand the phenomena. Although this may be beyond the scope of a revision, it is at least worth discussing if the findings might be specific to English. Reviewer 1 also raises concerns about the statistics that need to be clarified in addition to the several smaller comments from both reviewers.

Reviewer(s)' Comments to Author:

Referee: 1

Comments to the Author(s)

Summary and potential value of the contribution:

A fundamental property of human language is structural hierarchy. The evolution of hierarchical structure, and the degree to which there are analogs in non-human animals has been a subject of much research and debate. A challenge of such research, however, is that data from non-human animals is generally indirect (one cannot ask a humpback whale for grammaticality judgments). As such, statistical methods that track hierarchical structure may provide a useful way to directly compare humans to non-human animals.

The present paper investigates one such method, that looks at long-range dependencies between elements (using mutual information functions between elements of varying distances). For human language, long distance dependencies have been shown to follow a power law, which has been claimed to result from the hierarchical structure of human language. The contribution of the present paper is to throw this interpretation into question by showing that the same statistical pattern holds for productions of human infants that are assumed to be too young to have adult-like hierarchical structure.

General evaluation:

The general argument is interesting and potentially very useful to our understanding of the statistics of natural language. I nevertheless have some reservations with the paper in its present form. Of particular note, it currently lacks a positive proposal to explain the statistical patterns in question, which leaves the reader in a state of some mystery, not knowing how to interpret the results.

Main comments:

1. The paper would be significantly stronger if it could clearly and precisely outline a positive proposal -- a better model to explain the mechanisms by which such statistical patterns arise. If we are convinced by the arguments, then the patterns do not come from hierarchical organization of language. But then, what causes them? This being said, it is of course scientifically important to refute incorrect hypotheses. And the authors may decide that it is out of the scope of the present paper to provide a positive proposal to explain the facts. In this case, the degree to which the paper is publishable may depend on the degree to which the explanation in terms of hierarchy is widely assumed conventional wisdom. I have a hard time assessing this latter point.

2. I am not sure that what is being called hierarchy in this paper corresponds to the syntactic and phonological hierarchy that linguists have shown to exist for natural human language.

I am particularly confused by the results shown in Figure 3D. The authors show that shuffling phones within the utterance nevertheless allows long-range information (governed by the power law) to persist. The authors argue that this is problematic for the hierarchical interpretation of the pattern because it is showing up in children too young to have acquired complex linguistic structure. But the problem goes deeper than this. If similar results held for *adults*, it would also be an argument against interpreting this as linguistic hierarchy in any standard way. Specifically, the combinatorial systems of phonology and syntax are generally held to be two separate modules that largely do not interact (see, e.g. Pullum & Zwicky 1988; also relevant is the concept of "duality of patterning"). Thus, by scrambling the two systems in Figure 3D, both systems are completely disrupted. The only kind of "structure" remaining is discourse structure. But while there may be rhetorical relations that maintain discourse coherence, these are not generally taken to be the the same kind of linguistic structure that is arguably specific to humans (cf. Bolhuis et al. 2014).

Stated very coarsely, what the data in Figure 3D seem to show is that if one uses the sound /p/ in one utterance, one is more likely to use the sound /p/ in the following utterance. I don't believe that this follows from any principle of human phonology or syntax -- for infants or for adults.

3. Methodology: there is something paradoxical in the methods. The authors claim that children at 6-12 months are still in the babbling stage of development, so have no linguistic structure. But in Figure 3, the authors show that they shuffled words within the transcript, utterances within the transcript, and phones within the utterance -- even for children 6-12 months. But this presupposes that it is possible to identify words and utterances for children 6-12 months old. In other words, the authors identify three levels of hierarchy while at the same time asserting that hierarchy does not yet exist for this age group.

4. It could be useful to compare to other statistical methods that have been argued to provide evidence for syntactic combination. See, e.g. Yang (2013) for another way of characterizing the statistical profile of grammatical rules. Applying these methods to the sets of data analyzed here could provide stronger evidence for the (reasonable) assumption in the present paper that young children have not yet acquired complex linguistic structure. Ideally, what you would expect to find would be that the Yang measure becomes more adult-like over time, but that MI is adult-like from the beginning.

5. This paper should ideally be reviewed by someone who can check the statistical methods in more detail. I have just a couple of comments:

-Formula (1): shouldn't there be hats on the S's? S without a hat is never defined.

-Formula (2): for the distribution [a, b, c], I suppose that $N=3$ and $K=3$. N_i is not defined.

-It could be useful to provide simple examples of high and low entropy to give an intuition for how MI models long-distance dependency.

-I did not follow why "permuting X and Y [a]ffects the joint entropy ... but not the marginal entropies." If a simple example can be given briefly, this might be helpful.

Based on the shuffling manipulations, the paper makes the following generalization: "Across each shuffle analysis, we observe that short-range information content captured by exponential decay is largely captured within words and utterances, while long-range information is carried between utterances, even during early language production." While the graphs look somewhat different, I was unsure what the statistical basis for this conclusion was, given that the best model in many cases was still the combined `pow_exp` model.

References

Bolhuis, Johan J., Ian Tattersall, Noam Chomsky, and Robert C. Berwick. 2014. How could language have evolved? *PLoS Biology* 12.e1001934.

Pullum, G. K., & Zwicky, A. M. (1988). The syntax-phonology interface. *Linguistics: The Cambridge Survey: Volume 1, Linguistic Theory: Foundations*, 1, 255.

Yang, Charles. (2013). Ontology and phylogeny of language. *PNAS*, 110(16): 6324-6327.

Referee: 2

Comments to the Author(s)

General comments: Overall, this manuscript is clear, well-written, and interesting to read. The authors build off of their previous work in adult human speech and animal communication, and contextualize their research in the broader topic of sequential behaviors. The authors provide compelling evidence for long-range dependencies and hierarchical structure in the vocalizations of developing children that are reminiscent of those observed in adult speech.

Major comments:

While I think their findings are very interesting, I'd like the authors to speak more about the novelty of the work. Their results suggest that adult-like hierarchical structure can be observed even in the simple grammars of children: the utterances of children have some repeatable and predictable sequence structure, therefore can be described by Markovian and/or power-law decay functions. The authors are correct to highlight that hierarchical structure is also observed in non-linguistic human behaviors as well as in the vocalizations of non-human animals. So, how novel/unexpected is it that the speech patterns of children, even infants and toddlers, contain hierarchical structure?

The authors should discuss how the parameters of the power-law, exponential decay, and composite models compare between the current dataset and the corpus of adult speech from Sainburg et al. (2019), as well as age-dependent variation in the distance (in phones) at which the MI decay transitions from exponential to power law. I can appreciate the complexity of such comparisons across different levels of speech, but these comparisons would allow readers to assess how "adult-like" these long-range statistical relationships are in developing children.

The authors write that "long-range dependencies in language and other human behavior [33, 60, 61] may reflect more general biological processes inherited from the organization of underlying neurophysiological mechanisms [62–65]." In order to distinguish the contribution of linguistic input (learning) from fundamental ("innate") biological contributions, it would be revealing to compare the speech of at least one group of children that speak a different language. For example, because the shape of the MI functions seem to differ between adult English, German, and Japanese-speaking adults (Sainburg et al., 2019), comparing the English-speaking children to Japanese- or German-speaking children would be very informative. If the shape of the curve in children is a function of biology and not linguistic exposure, then the shapes should be statistically indistinguishable across children that speak different languages with different MI functions.

Minor comments:

In the last sentence of the abstract, the authors write that "(t)hese linguistic structures cannot, therefore, be the sole cause of long-range statistical dependencies in language." It is not really clear about the degree to which the authors are claiming that these long-range dependencies in infant vocalizations are innate. One question is to what degree long-range dependencies can be inferred from auditory patterns heard in utero. There is increasing evidence that auditory experiences in utero shape the brain and vocal output in humans, and it would be useful for the authors to discuss the extent to which in utero experiences could allow infants and toddlers to communicate with hierarchical structure.

Line 182-185: The authors should provide information about how prevalent repeated elements are at different levels (e.g., words, utterances) and different ages. This would provide some insight into the importance of this set of analyses.

Author's Response to Decision Letter for (RSPB-2021-0806.R0)

See Appendix A.

RSPB-2021-2657.R0

Review form: Reviewer 1

Recommendation

Accept with minor revision (please list in comments)

Scientific importance: Is the manuscript an original and important contribution to its field?

Excellent

General interest: Is the paper of sufficient general interest?

Excellent

Quality of the paper: Is the overall quality of the paper suitable?

Excellent

Is the length of the paper justified?

Yes

Should the paper be seen by a specialist statistical reviewer?

Yes

Do you have any concerns about statistical analyses in this paper? If so, please specify them explicitly in your report.

Yes

It is a condition of publication that authors make their supporting data, code and materials available - either as supplementary material or hosted in an external repository. Please rate, if applicable, the supporting data on the following criteria.

Is it accessible?

Yes

Is it clear?

Yes

Is it adequate?

Yes

Do you have any ethical concerns with this paper?

No

Comments to the Author

Summary:

Human language is known for displaying hierarchical structure. The relation between linguistic hierarchy and other hierarchical organization in human and non-human behavior is a rich area of recent research. The present paper investigates one recent statistical method that has been used to detect and compare hierarchical structure for human and non-human animals (a power-law decay of long-distance dependencies). The results of the paper provide important new insights into how to best interpret these statistical findings (namely, they must be explained at least in part by organizational mechanisms outside of linguistic hierarchy).

Evaluation:

I reviewed a previous version of this paper. Overall, I find the new version of the paper to be greatly improved. The newly added paragraphs in the discussion section adequately address my previous concerns regarding the lack of a positive proposal; the discussion section now provides several different explanations of the data that could be investigated in further work. The new subsection in the introduction on different kinds of hierarchy is useful for framing the goals of the paper. The comparison of the shuffle procedure to the same procedure on an idealized model is useful for understanding the effects found.

I have a few remaining concerns:

First, I found it rather difficult to understand the goals of the newly added sections 2.2 and 3.1. The conclusion from section 3.1 is that "across all age groups, transcripts are more hierarchically compressible than their Markov or randomized counterparts" and that this is "similar across each age group." Is this then another way (beyond the MI analysis) of showing that hierarchical organization exists before the acquisition of linguistic structure? If so, then what does the MI analysis show beyond what we'd expect from the results using Sequitur hierarchical compressibility? The paper would be improved if the authors could more clearly lay out how these two analyses relate to each other, and what insights each gives about the hierarchical structure of the data.

Second, in section 3.3, comparison to an ideal model provides a very useful visual baseline for understanding what is going on in the shuffle analyses, but I think it bears noting that this section of the paper does not report any statistical tests. I'm not sure how problematic this is for the paper, given that there are statistical tests reported for the main results in section 3.2 (Tables S1-S2). But this feels like a somewhat important gap. (Note also that statistical results are not presented in section 3.1 but probably could be.)

Finally, the idealized model in Section 3.3 suggests that phonological relations are Markovian but that syntactic relations are hierarchical. If this is true, then it is perhaps somewhat surprising that there is still a significant exponential component to the MI functions in the analysis based on word distance (Figures 3DEF). This is potentially related to the finding that, when phones are shuffled, the best model still contains an exponential component (Figure 4I). Some discussion of this point could potentially be clarifying.

Minor comments:

Line 197: is "interred" the intended verb here?

Figure 2: I'm not sure how useful the example Markov models are. They may be too messy to provide any meaningful information.

Figure 3: It is currently confusing why the 6-12 month age group is excluded for the MI analyses for word distance, given that words are coded for this age group (as seen in the shuffle analysis). I think ultimately the answer to this is rather mundane: namely, the analysis of word distance used CHILDES data instead of Phonebank data, and CHILDES data starts at 12 months. Perhaps this confusion could be avoided by mentioning this fact explicitly in the caption of the Figure. (Lines 285-286 similarly implicate that the paper *failed* to find long-range statistical relationships in word sequences. It just wasn't tested.)

Line 225: Table S3, I think, not Table 3.

Figure 4:

-Please move the age-group labels ("6-12 months", etc.) to a more noticeable place, possibly below the last graphs.

-The labels I and H should be swapped to respect alphabetical order.

-The labels used in the caption currently do not correspond to the labels of the figures.

-It should be clarified that when the exponential model is the best fit, the exponential component (the only component) is a solid line and the power-law component is not shown.

Line 232-233: "the PCFG model from Figure 1". Do the paper or supplemental materials ever provide the precise PCFG that was used to generate the graphs in Figure 1 Figure 4BCDE?

Review form: Reviewer 2

Recommendation

Accept with minor revision (please list in comments)

Scientific importance: Is the manuscript an original and important contribution to its field?

Excellent

General interest: Is the paper of sufficient general interest?

Excellent

Quality of the paper: Is the overall quality of the paper suitable?

Excellent

Is the length of the paper justified?

Yes

Should the paper be seen by a specialist statistical reviewer?

No

Do you have any concerns about statistical analyses in this paper? If so, please specify them explicitly in your report.

No

It is a condition of publication that authors make their supporting data, code and materials available - either as supplementary material or hosted in an external repository. Please rate, if applicable, the supporting data on the following criteria.

Is it accessible?

Yes

Is it clear?

Yes

Is it adequate?

Yes

Do you have any ethical concerns with this paper?

No

Comments to the Author

GENERAL COMMENTS: The revised manuscript clarifies a number of important aspects outlined by the reviewers. I appreciate the additional analyses and points in the Discussion section. In my opinion, the manuscript remains an excellent contribution to our understanding of the sequential and hierarchical organization of behavior. I have some outstanding issues that should be clarified before the article is published.

I like the addition of French in the analysis as an additional data point for children but, without knowing the hierarchical structure of adult French, it is not clear if this analysis satisfied my concern. I understand that their analyses are limited to the availability of annotated corpora, but I was hoping that the authors could analyze the developmental pattern of hierarchical structures for languages that have MI decay functions that are distinct from adult English (e.g., adult German based on figures in Sainburg et al., 2019). It would be interesting to see if children raised with languages that have different MI decay functions (as determined by the analysis of adult speech) also demonstrate different MI decay functions (or are organizational differences only found in adult speech). Have the authors analyzed the sequential organization of adult French in this manner? If so, does the hierarchical structure of adult French differ from adult English?

In the previous round, I recommended that the authors discuss how “adult-like” the organizational patterns of children speech was by plotting the data for adults in the same figure as for children (either in the main text or supplementary information section). A direct comparison of this nature is important to understand how the temporal patterning of communication changes over time, to think about mechanisms of behavioral control and organization, and to understand what the authors exactly mean by “adult-like”. In addressing my suggestion, the authors discuss the complexities of such an analysis given differences in transcript sizes across corpora and the effects of transcript size on parameter estimates (e.g., Figure S3). However, isn’t it possible to analyze adult corpora that have been filtered or pruned in such a manner to be equivalent in transcript sizes as those for children? Or to run an analysis between adult and children where transcript size is used as a covariate?

MINOR POINTS:

Figure 2A-C. The authors should provide more information on this plot. How should one interpret the position of circles and distance of circles from each other (if the dimensions and distances meaningful at all). I assume the thickness/darkness of the arrows indicates transition probability but that is not mentioned in the figure legend. While this graphical depiction is more interesting than a table, readers might be more used to seeing a table with transition probabilities between phones to summarize first-order transitions.

Line 248-251: I realize that these sentences refer to different panels in Figure 4 but the sentences overlap so much that it’s hard to differentiate. Maybe combine into one sentence? Or maybe re-write in a manner that allows readers to better differentiate.

Line 251-253: indicate what is the parallel for this type of shuffling? (e.g., like the authors did in lines 247-248: Shuffling Markov sequences within the transcript (Figure 4C) relates to shuffling words within the transcript (4G).”)

Figure 4 (and Figures S4, S5). It is sometimes hard to reconcile the main text with indications of best model fits above the panels. For example, in lines 254-255, it is mentioned that shuffling phones within utterances reduces the exponential component; consequently, one might intuit that

the best function would be simply the power law function. But in Figure 4I, the power-exponential model provides the best fit for each age group. (I understand that the exponential component is reduced, not necessarily eliminated.). So, I suggest the authors add additional information to the panels (instead of just listing the best model), maybe the AIC values for each of the different models. (I acknowledge that this information is in Tables 1 & 2.). Basically, I would like the information in the figure to match the text better.

Line 285: not clear what "adult-like" means. Adding a few sentences describing the structure of adult speech at the beginning of the Discussion would be useful.

Line 306-307: the wording here should be more consistent with the analysis. For example, the authors could write: "Although including information about the extent of adult speech with children in the corpora did not increase explanatory power, ...". The current wording is too vague and subject to over-interpretation.

Figure S2. Add age groups to the tops of each panel. Better yet, why not replot the figures for English above or below the plots for French.

Figure S4 vs. S5. When you shuffle between utterances in Figure S4, the power-exponential model provides the best fit for all groups. However, when you shuffle between utterances in Figure S5, there is more variability in the best fit. Please discuss.

Figure S7 and S8 are not cited in the main text, though maybe S7 and S8 should be references on line 265?

Decision letter (RSPB-2021-2657.R0)

24-Jan-2022

Dear Mr Sainburg

I am pleased to inform you that your manuscript RSPB-2021-2657 entitled "Long-range sequential dependencies precede complex syntactic production in language acquisition" has been accepted for publication in Proceedings B.

The referee(s) have recommended publication, but also suggest some minor revisions to your manuscript. Therefore, I invite you to respond to the referee(s)' comments and revise your manuscript. Because the schedule for publication is very tight, it is a condition of publication that you submit the revised version of your manuscript within 7 days. If you do not think you will be able to meet this date please let us know.

Sincerely,
Dr Robert Barton
mailto: proceedingsb@royalsociety.org

Associate Editor

Comments to Author:

Thank you for the careful revision of the manuscript. Both the reviewers and I find the manuscript greatly improved. However, both reviewers still see room for additional improvement and clarification. Reviewer 1 suggests the French analysis would be more useful if it could be shown that French has a different MI decay structure than English. If this information is not readily available, it would be good to discuss this limitation. Reviewer 1 also suggests ways to directly compare the organizational structure of adult and children's speech. Both reviewers suggest improving or (or possibly removing) Figure 2 A-C or, at least, improving the description of it. Both reviewers make several other suggestions for clarification.

Reviewer(s)' Comments to Author:

Referee: 2

Comments to the Author(s).

GENERAL COMMENTS: The revised manuscript clarifies a number of important aspects outlined by the reviewers. I appreciate the additional analyses and points in the Discussion section. In my opinion, the manuscript remains an excellent contribution to our understanding of the sequential and hierarchical organization of behavior. I have some outstanding issues that should be clarified before the article is published.

I like the addition of French in the analysis as an additional data point for children but, without knowing the hierarchical structure of adult French, it is not clear if this analysis satisfied my concern. I understand that their analyses are limited to the availability of annotated corpora, but I was hoping that the authors could analyze the developmental pattern of hierarchical structures for languages that have MI decay functions that are distinct from adult English (e.g., adult German based on figures in Sainburg et al., 2019). It would be interesting to see if children raised with languages that have different MI decay functions (as determined by the analysis of adult speech) also demonstrate different MI decay functions (or are organizational differences only found in adult speech). Have the authors analyzed the sequential organization of adult French in this manner? If so, does the hierarchical structure of adult French differ from adult English?

In the previous round, I recommended that the authors discuss how "adult-like" the organizational patterns of children speech was by plotting the data for adults in the same figure as for children (either in the main text or supplementary information section). A direct comparison of this nature is important to understand how the temporal patterning of communication changes over time, to think about mechanisms of behavioral control and organization, and to understand what the authors exactly mean by "adult-like". In addressing my suggestion, the authors discuss the complexities of such an analysis given differences in transcript sizes across corpora and the effects of transcript size on parameter estimates (e.g., Figure S3). However, isn't it possible to analyze adult corpora that have been filtered or pruned in such a manner to be equivalent in transcript sizes as those for children? Or to run an analysis between adult and children where transcript size is used as a covariate?

MINOR POINTS:

Figure 2A-C. The authors should provide more information on this plot. How should one interpret the position of circles and distance of circles from each other (if the dimensions and

distances meaningful at all). I assume the thickness/darkness of the arrows indicates transition probability but that is not mentioned in the figure legend. While this graphical depiction is more interesting than a table, readers might be more used to seeing a table with transition probabilities between phones to summarize first-order transitions.

Line 248-251: I realize that these sentences refer to different panels in Figure 4 but the sentences overlap so much that it's hard to differentiate. Maybe combine into one sentence? Or maybe re-write in a manner that allows readers to better differentiate.

Line 251-253: indicate what is the parallel for this type of shuffling? (e.g., like the authors did in lines 247-248: Shuffling Markov sequences within the transcript (Figure 4C) relates to shuffling words within the transcript (4G).")

Figure 4 (and Figures S4, S5). It is sometimes hard to reconcile the main text with indications of best model fits above the panels. For example, in lines 254-255, it is mentioned that shuffling phones within utterances reduces the exponential component; consequently, one might intuit that the best function would be simply the power law function. But in Figure 4I, the power-exponential model provides the best fit for each age group. (I understand that the exponential component is reduced, not necessarily eliminated.). So, I suggest the authors add additional information to the panels (instead of just listing the best model), maybe the AIC values for each of the different models. (I acknowledge that this information is in Tables 1 & 2.). Basically, I would like the information in the figure to match the text better.

Line 285: not clear what "adult-like" means. Adding a few sentences describing the structure of adult speech at the beginning of the Discussion would be useful.

Line 306-307: the wording here should be more consistent with the analysis. For example, the authors could write: "Although including information about the extent of adult speech with children in the corpora did not increase explanatory power, ...". The current wording is too vague and subject to over-interpretation.

Figure S2. Add age groups to the tops of each panel. Better yet, why not replot the figures for English above or below the plots for French.

Figure S4 vs. S5. When you shuffle between utterances in Figure S4, the power-exponential model provides the best fit for all groups. However, when you shuffle between utterances in Figure S5, there is more variability in the best fit. Please discuss.

Figure S7 and S8 are not cited in the main text, though maybe S7 and S8 should be references on line 265?

Referee: 1

Comments to the Author(s).

Summary:

Human language is known for displaying hierarchical structure. The relation between linguistic hierarchy and other hierarchical organization in human and non-human behavior is a rich area of recent research. The present paper investigates one recent statistical method that has been used to detect and compare hierarchical structure for human and non-human animals (a power-law decay of long-distance dependencies). The results of the paper provide important new insights into how to best interpret these statistical findings (namely, they must be explained at least in part by organizational mechanisms outside of linguistic hierarchy).

Evaluation:

I reviewed a previous version of this paper. Overall, I find the new version of the paper to be greatly improved. The newly added paragraphs in the discussion section adequately address my previous concerns regarding the lack of a positive proposal; the discussion section now provides several different explanations of the data that could be investigated in further work. The new subsection in the introduction on different kinds of hierarchy is useful for framing the goals of the paper. The comparison of the shuffle procedure to the same procedure on an idealized model is useful for understanding the effects found.

I have a few remaining concerns:

First, I found it rather difficult to understand the goals of the newly added sections 2.2 and 3.1. The conclusion from section 3.1 is that "across all age groups, transcripts are more hierarchically compressible than their Markov or randomized counterparts" and that this is "similar across each age group." Is this then another way (beyond the MI analysis) of showing that hierarchical organization exists before the acquisition of linguistic structure? If so, then what does the MI analysis show beyond what we'd expect from the results using Sequitur hierarchical compressability? The paper would be improved if the authors could more clearly lay out how these two analyses relate to each other, and what insights each gives about the hierarchical structure of the data.

Second, in section 3.3, comparison to an ideal model provides a very useful visual baseline for understanding what is going on in the shuffle analyses, but I think it bears noting that this section of the paper does not report any statistical tests. I'm not sure how problematic this is for the paper, given that there are statistical tests reported for the main results in section 3.2 (Tables S1-S2). But this feels like a somewhat important gap. (Note also that statistical results are not presented in section 3.1 but probably could be.)

Finally, the idealized model in Section 3.3 suggests that phonological relations are Markovian but that syntactic relations are hierarchical. If this is true, then it is perhaps somewhat surprising that there is still a significant exponential component to the MI functions in the analysis based on word distance (Figures 3DEF). This is potentially related to the finding that, when phones are shuffled, the best model still contains an exponential component (Figure 4I). Some discussion of this point could potentially be clarifying.

Minor comments:

Line 197: is "interred" the intended verb here?

Figure 2: I'm not sure how useful the example Markov models are. They may be too messy to provide any meaningful information.

Figure 3: It is currently confusing why the 6-12 month age group is excluded for the MI analyses for word distance, given that words are coded for this age group (as seen in the shuffle analysis). I think ultimately the answer to this is rather mundane: namely, the analysis of word distance used CHILDES data instead of Phonebank data, and CHILDES data starts at 12 months. Perhaps this confusion could be avoided by mentioning this fact explicitly in the caption of the Figure. (Lines 285-286 similarly implicate that the paper *failed* to find long-range statistical relationships in word sequences. It just wasn't tested.)

Line 225: Table S3, I think, not Table 3.

Figure 4:

- Please move the age-group labels ("6-12 months", etc.) to a more noticeable place, possibly below the last graphs.
- The labels I and H should be swapped to respect alphabetical order.
- The labels used in the caption currently do not correspond to the labels of the figures.

-It should be clarified that when the exponential model is the best fit, the exponential component (the only component) is a solid line and the power-law component is not shown.

Line 232-233: "the PCFG model from Figure 1". Do the paper or supplemental materials ever provide the precise PCFG that was used to generate the graphs in Figure 1 Figure 4BCDE?

Author's Response to Decision Letter for (RSPB-2021-2657.R0)

See Appendix B.

Decision letter (RSPB-2021-2657.R1)

01-Feb-2022

Dear Mr Sainburg

I am pleased to inform you that your manuscript entitled "Long-range sequential dependencies precede complex syntactic production in language acquisition" has been accepted for publication in Proceedings B.

Your article has been estimated as being 9 pages long. Our Production Office will be able to confirm the exact length at proof stage.

Data Accessibility section

Open Access

Paper charges

Sincerely,
Editor, Proceedings B
<mailto:proceedingsb@royalsociety.org>

Appendix A

Dear Dr. Barton,

Thank you very much for the opportunity to revise our manuscript, titled *Long-range sequential dependencies precede complex syntactic production in language acquisition* (RSPB-2021-0806). We greatly appreciate the constructive comments from each of the reviewers and have revised the manuscript extensively in light of their suggestions. Major revisions, reflecting the concerns of multiple reviewers, are described below. Following this overview, we provide a point-by-point response to each of the reviewers' questions and concerns. We hope you agree that the end result is a much-improved paper that is now ready for publication.

In the submitted manuscript substantial changes to the text are given in **purple**.

Major changes

- There were several questions about the shuffling controls (comments 0.1, 0.2, 1.2, 1.3). To address these remarks, we now model our hypotheses from Figure 1 in Figure 3 to explicitly relate our hypotheses directly to our empirical results.
- We added a comparison to a second language dataset (French) as requested in comment 2.3.
- We added two new statistical analyses of our datasets. These provide context and help explain our main mutual information analysis, as requested in comment 1.4:
 - an analysis exhibiting the non-random transition structure in each age group
 - an analysis demonstrating the hierarchical compressibility of each dataset
- We added an important discussion of the distinctions between general hierarchical organization, linguistic hierarchical organization (e.g. syntactic, phonological, discourse hierarchy), and hierarchical description to address comments 1.1 and 1.2.
- We added an analysis of model fit parameters to address comment 2.2.

Sincerely,

Tim Sainburg, Anna Mai, Tim Gentner

Point by point responses to verbatim concerns (**shown in blue**)

Editor remarks to Author:

This manuscript was reviewed by two experts and myself. We all found the manuscript to be interesting and potentially worthy of publication as it addresses important issues about language development. However, both reviewers raised serious concerns that prevent it from being published in its current form. It is not clear at this stage if the concerns can be adequately addressed. If they can be addressed, I would allow resubmission and attempt to have the manuscript reviewed by the same reviewers.

- 0.1. Reviewer 1 points out a potential problem with the methods. How are there 3 levels of hierarchy in the speech of 6-12 month old children which are needed to perform the shuffling analyses shown in figure 3? A central claim is that these children lack these structures.

The shuffling analysis in the original figure 3 (revised figure 4) relies on a hierarchical description of human speech taken directly from the published transcription formats (MacWhinney, 2000) used for the CHILDES and PHONBANK datasets, in which utterances comprise words, which comprise phones. Beyond this technical fact, however, the reviewer raises an important point regarding differences between *linguistic hierarchy*, *general hierarchy*, and a *hierarchical description*. These ways of discussing hierarchy are often conflated in papers (and in our initial manuscript). We have worked to do a better job of distinguishing them in the revision. Briefly, hierarchical organization in signals may be the result of linguistic or more general (non-linguistic) production processes (or both), and even signals that are not produced by hierarchical processes may still be described (e.g. compressively represented) hierarchically (Gomez-Martin et al., 2016; Figure 3a). To clarify these important differences in the use of the term ‘hierarchy’, we have added a section to the paper (as requested in 1.2) distinguishing between linguistic hierarchy, general hierarchy, and hierarchical description (Section *General hierarchical organization, linguistic hierarchy, and hierarchical description* in the main text). Section 3.1 of the revised manuscript and the accompanying Figure 2 also touch on these ideas, as do sections 1.1 and 1.2.1 of this rebuttal.

Our central claim is that long-range relationships exist in young children’s vocalizations despite their lack of complex hierarchical linguistic structure of the kind assumed to underpin adult language syntax. The notion that children lack this kind of hierarchical linguistic structure isn’t controversial. Reviewer 1 (see 1.4) agrees that this position is reasonable but asked for supporting evidence using our datasets, suggesting we repeat the analysis by Yang (2013) on the CHILDES dataset. We note that a very similar analysis on these same datasets has already been performed (Meylan et al., 2017), the results of which support our premise that young children’s vocalizations do not have complex linguistic structure. Additionally, we added references to several other works reviewing the literature on the developmental time course of hierarchical linguistic structure (discourse and syntactic structure) in vocal productions.

- 0.2. Another problem raised by Reviewer 1 is that it is not clear what the shuffling of phones analysis (shown in 3D) is doing and how it might relate to the different hierarchies present in adult speech. Repeating these analyses with adult speech or at least better integrating the work with other descriptions of adult speech (as suggested by Reviewer 2) and other methods of characterizing speech (as suggested by Reviewer 1) would help clarify the issue. This might allow you to better differentiate whether the long-range dependencies found here are stepping stones towards hierarchical speech (i.e., ‘precede’ adult speech) or are something unrelated.

To aid the reader in understanding the shuffles and how they might relate to different hierarchies present in adult speech, we reformatted the original figure 3 (now Figure 4), adding an explicit model of our hypotheses to demonstrate the expected effect of shuffling on each hierarchical level.

Our dataset has three levels of hierarchy (utterance, word, and phone; Fig 4F) and we shuffle independently at each level. To help the reader understand the effects of a shuffle at one level on other levels, we apply each shuffle to synthetic sequences generated from the Probabilistic Finite State Grammar with Markov-generated endpoints (see Figure 1 and Sainburg et. al., 2019), then

describe the pattern of mutual information decay (Figure 4 C-E) in the resulting sequence. The synthetic data help explain the shuffles of real data (Figure 4 G-I).

To understand what happens at higher levels when we shuffle the speech corpora at the phone level (i.e., “phones within utterance”), one can look at the synthetic data where we shuffle “within Markov sequence”. This shuffle removes most of the exponential decay from the sequence while preserving power-law decay (Figure 4D). The same effect is observed in the real data when we shuffle “phones within utterance”(see Figure 4I), and is consistent with the notion that phonological relationships within adult speech follow Markovian dynamics (Kaplan & Kay, 1994).

Results with both the synthetic and the real datasets show that residual sequence-structure persists at un-shuffled levels in the hierarchy. In the synthetic data, the structure at different levels is independent, and the correspondence between the synthetic and real data suggests at least some independence across levels in the latter. We had already performed the same shuffling experiments on adult speech (Sainburg et al., 2019, Supplementary Figure 2), as suggested, and observed similar results. For this reason, we are reluctant to claim that different levels “precede” others in a dependent manner. Our intent in using this term was only to imply a temporal ordering over the course of development. Because we are only describing observed structure, we think it’s best to remain agnostic on any dependencies between processes that shape structure at different hierarchical levels.

- 0.3. In addition, there is general confusion about how long-range dependency might relate to hierarchical structures. Is one necessary for the other? This relates to Reviewer 1’s suggestion to include a positive explanation for what is happening--how can we get long-range dependency without hierarchical structure (e.g., is, as suggested by the example in figure 3, it just a matter of continuing to discuss the same topic which can lead to repeating phones over long periods of time?).

We agree that clarity on the relationships between long-range dependencies and hierarchical structure is crucial to understanding our work. We rewrote several sections of the manuscript with this clarity in mind (in particular the section on general hierarchy in the introduction and paragraph two of the discussion). Although they often co-occur, long-range dependencies and hierarchical structure are not the same. Long-range dependencies can be produced by hierarchical and non-hierarchical processes.

The editor mentions that continuing to discuss the same topic can lead to repeating phones over long periods of time. Indeed, Altmann et al (2006), show that the long-range correlations in text attributed to hierarchical discourse structure flow between linguistic levels of organization: because at the highest level in the hierarchy topics control the distribution of words within them, some words and the phonemes they are composed of are more likely, and therefore are sampled with a greater probability. Thus, more frequent sampling of a word (i.e. repetition) within a topic is a hierarchical source of long-range dependencies, because the probability distribution of phonemes changes depending on the topic that is currently being discussed.

Repetition can also cause long-range correlations that are not attributed to hierarchy. For example, Markov renewal processes can produce long-range correlations in Markov processes by emitting n repeats of the current state from a non-Markov-sampled distribution (Kershenbaum et al., 2014).

We control for direct repetitions (as in Markov renewal processes) in Figure S4, though more complex forms of repetitions such as repetitions of looping patterns would be non-trivial to discover and control for (and would themselves be hierarchical).

We address Reviewer 1's specific suggestion for positive explanation in the context of their remarks (Section 1.1 and 1.2),

- 0.4. Reviewer 2 also points out that looking at this in another language might help understand the phenomena. Although this may be beyond the scope of a revision, it is at least worth discussing if the findings might be specific to English.

We agree that comparing the information decay parameters across ages in different languages could provide an interesting and important extension to understanding our results. To this end, we added a second language dataset, French, which had a number of datasets of similar ages and transcript lengths readily available in the PhonBank database. We found the same decay results in this dataset as we did in English, which we added as a supplementary figure (Supplementary Figure 2).

We then computed decay parameters for both English and French across ages both across transcripts and within the longest transcripts in each dataset. Broadly, we found similar changes in each of the decay parameters across these two languages (Supplementary Figure 3). We do not believe it is prudent to draw any strong conclusions from quantitative differences in parameters of the decay fits, however, as they are largely dependent upon the size of the transcripts available, which tend to be much longer for older age groups (Supplementary Figure 1). Still, it could be possible to artificially control the dataset sizes from older age groups, as well as extrapolate parameter trends by fitting a model of parameter change over time to each transcript / age group and comparing the parameter change of the model (of the model) as a function of age. If the reviewer thinks this analysis is necessary, we would be happy to provide it.

- 0.5. Reviewer 1 also raises concerns about the statistics that need to be clarified in addition to the several smaller comments from both reviewers.

We address these concerns in the context of each specific critique.

1. Reviewer 1

Summary and potential value of the contribution:

A fundamental property of human language is structural hierarchy. The evolution of hierarchical structure, and the degree to which there are analogs in non-human animals has been a subject of

much research and debate. A challenge of such research, however, is that data from non-human animals is generally indirect (one cannot ask a humpback whale for grammaticality judgments). As such, statistical methods that track hierarchical structure may provide a useful way to directly compare humans to non-human animals.

The present paper investigates one such method, that looks at long-range dependencies between elements (using mutual information functions between elements of varying distances). For human language, long distance dependencies have been shown to follow a power law, which has been claimed to result from the hierarchical structure of human language. The contribution of the present paper is to throw this interpretation into question by showing that the same statistical pattern holds for productions of human infants that are assumed to be too young to have adult-like hierarchical structure.

General evaluation:

The general argument is interesting and potentially very useful to our understanding of the statistics of natural language. I nevertheless have some reservations with the paper in its present form. Of particular note, it currently lacks a positive proposal to explain the statistical patterns in question, which leaves the reader in a state of some mystery, not knowing how to interpret the results.

Main comments:

- 1.1. The paper would be significantly stronger if it could clearly and precisely outline a positive proposal -- a better model to explain the mechanisms by which such statistical patterns arise. If we are convinced by the arguments, then the patterns do not come from hierarchical organization of language. But then, what causes them? This being said, it is of course scientifically important to refute incorrect hypotheses. And the authors may decide that it is out of the scope of the present paper to provide a positive proposal to explain the facts. In this case, the degree to which the paper is publishable may depend on the degree to which the explanation in terms of hierarchy is widely assumed conventional wisdom. I have a hard time assessing this latter point.

The goal of the present paper is to examine the statistical structure in children's speech. Having observed a particular pattern, we then seek to rigorously support our observation that the pattern exists in the absence of adult linguistic structure. We believe that we have done this, and that our observations are reliable, novel, and noteworthy in their own right, pointing to the likelihood that hierarchical but non-linguistic sources shape power-law dependencies in young children's speech.

We agree with the reviewer that the cause (or causes) of the observed patterns are of great interest, but respectfully submit that a rigorous test of any hypothetical cause is no small undertaking and well beyond the scope of the present paper (as the reviewer intuitively). We have of course considered possible non-linguistic causes and now note several potential mechanisms in the discussion (lines 293-3337), including those that may be non-hierarchical but which may still produce a power-law decay (whether the latter mechanisms even exist is a topic of ongoing debate (e.g. Munoz, 2018; Gisiger 2001)).

We also note that it is not our intention to suggest that a linguistic account of hierarchical structure in speech is necessarily wrong, and have revised the paper to avoid this suggestion. In the text, we state:

This does not rule out the possibility that long-range dependencies in adult language are driven in part by linguistic structures, but the absence of these organizational components in the youngest children indicates that other mechanisms likely also shape the long-range structure of speech.

Surely, the hierarchical structure of language gives rise to long-range dependencies as soon as more complex linguistic structures emerge in a speaker's repertoire - we do not refute this. But in young children, who lack complex linguistic structure (e.g. syntax or discourse structure, and see response section 1.4), such an account is incomplete. Thus, we present a novel finding, that long-range statistical dependencies are present in children's speech, and a novel implication, that the long-range dependencies present in speech cannot entirely be explained by linguistic structure. We believe that both of these points are important and worthy of publication.

1.2. I am not sure that what is being called hierarchy in this paper corresponds to the syntactic and phonological hierarchy that linguists have shown to exist for natural human language.

I am particularly confused by the results shown in Figure 3D. The authors show that shuffling phones within the utterance nevertheless allows long-range information (governed by the power law) to persist. The authors argue that this is problematic for the hierarchical interpretation of the pattern because it is showing up in children too young to have acquired complex linguistic structure. But the problem goes deeper than this. If similar results held for *adults*, it would also be an argument against interpreting this as linguistic hierarchy in any standard way. Specifically, the combinatorial systems of phonology and syntax are generally held to be two separate modules that largely do not interact (see, e.g. Pullum & Zwicky 1988; also relevant is the concept of "duality of patterning"). Thus, by scrambling the two systems in Figure 3D, both systems are completely disrupted. The only kind of "structure" remaining is discourse structure. But while there may be rhetorical relations that maintain discourse coherence, these are not generally taken to be the the same kind of linguistic structure that is arguably specific to humans (cf. Bolhuis et al. 2014).

Stated very coarsely, what the data in Figure 3D seem to show is that if one uses the sound /p/ in one utterance, one is more likely to use the sound /p/ in the following utterance. I don't believe that this follows from any principle of human phonology or syntax -- for infants or for adults.

The reviewer makes several points that we address one-by-one below.

1.2.1. I am not sure that what is being called hierarchy in this paper corresponds to the syntactic and phonological hierarchy that linguists have shown to exist for natural human language.

We agree that it is important to be more precise in our use of the term 'hierarchy' and to clarify the differences between *linguistic hierarchy*, *general hierarchy*, and a *hierarchical description*. We added a section to the paper distinguishing between these ideas (Introduction section *General*

hierarchical organization, linguistic hierarchy, and hierarchical description). Briefly, hierarchical organization in signals may be the result of linguistic or non-linguistic production processes (or both), and even signals that are not produced by hierarchical processes may still be described (e.g. compressively represented) hierarchically (e.g. Gomez-Martin et al., 2016; Figure 2). For the purposes of the analyses and shuffling controls in (revised Figures 3 and 4, respectively) we rely on a hierarchical description of human speech taken directly from the published transcription formats (MacWhinney, 2000) used for the CHILDES and PHONBANK datasets, in which utterances comprise words, which comprise phones. We do not explicitly manipulate syntactic organization, but assume that we affect it indirectly by manipulating word and utterance orders. In Figure 1, we present a model meant only to provide guidance in understanding how hierarchical and non-hierarchical (in this case Markovian) processes can combine, and how shuffling at various levels affects the dynamics of MI decay over increasingly longer intervals. To aid this interpretation, we added an analysis in which we generated synthetic sequences using this model, then shuffled these sequences (Figure 4B-E) at levels that we take to correspond to the shuffles of the real data (Figure 4G-I). The pattern of results in both data (real and synthetic) show a close correspondence (discussed in more detail in 1.2.2).

1.2.2. The authors show that shuffling phones within the utterance nevertheless allows long-range information (governed by the power law) to persist. The authors argue that this is problematic for the hierarchical interpretation of the pattern because it is showing up in children too young to have acquired complex linguistic structure. But the problem goes deeper than this. The authors show that shuffling phones within the utterance nevertheless allows long-range information (governed by the power law) to persist. [...] If similar results held for *adults*, it would also be an argument against interpreting this as linguistic hierarchy in any standard way. Specifically, the combinatorial systems of phonology and syntax are generally held to be two separate modules that largely do not interact (see, e.g. Pullum & Zwicky 1988; also relevant is the concept of "duality of patterning"). Thus, by scrambling the two systems in Figure 3D, both systems are completely disrupted. The only kind of "structure" remaining is discourse structure. But while there may be rhetorical relations that maintain discourse coherence, these are not generally taken to be the the same kind of linguistic structure that is arguably specific to humans (cf. Bolhuis et al. 2014)..

As is mentioned in 1.2.1, we added a shuffling analysis to Figure 4 over an explicit model in which finite-state (i.e. phonological) and hierarchical systems are completely separate modules, to aid in interpreting our shuffling results. We show that our shuffling results are fully consistent with the reviewer's account of separable modules. We additionally point to an equivalent shuffling analysis over adult speech in our previous paper (below), which is also consistent. To make this as clear as possible, a full section in the main text has been added (lines 232-261).

The purpose of shuffling phones within utterances is to alter the exponential component of the combined power-law/exponential model, based on the understanding that phonological sequencing is largely Markovian. Consistent with this understanding of phonology, the exponential component of the decay is lost with this shuffle (Figure 4D and I). While it is true that we show power-law governed long-range dependencies persist when phones are shuffled

within an utterance, this is to be expected, as we are not altering the probability distributions over longer distances, which is explicitly what the sequential mutual information is measuring.

We do not argue that this is problematic for the hierarchical interpretation of patterning. Instead, we are arguing against the notion that these relationships are driven by linguistic hierarchy because we observe them in children's speech, even when such linguistic structure is not present (we added lines 65-89 to make this more clear). We are not arguing against a general notion of hierarchy (as we mention in 1.1).

Analyses of adult speech are also consistent with this interpretation. We performed the same shuffling analysis on adult speech from two language corpora (English and Japanese) in a previously published paper (Sainburg et al., 2019, Supplementary Figure 2) and found similar power-law persistence (with some variation between languages). Again in adult speech, the source of this persistent structure is not clear, but it exists and is consistent with a hierarchical account (e.g. Figure 4a). In young children, the power law is not likely to be linguistic in origin. In adults, it may not be linguistic either, though we are not prepared to rule that out, as we mention in the discussion (line 293-300).

1.2.3. Stated very coarsely, what the data in Figure 3D seem to show is that if one uses the sound /p/ in one utterance, one is more likely to use the sound /p/ in the following utterance. I don't believe that this follows from any principle of human phonology or syntax -- for infants or for adults

This is almost exactly the case and is explained in some detail in Altmann et al., (2006) in discourse structure and Lin and Tegmark (2017) in syntax structure. To be more specific, however, the use of the sound /p/ in one sentence does *not* mean that the use of the sound /p/ is more likely in the following utterance, it instead implies that knowing the sound /p/ is in the current utterance, we have reduced the uncertainty in the distribution of phonemes in the following utterance.

To paraphrase Lin and Tegmark (2017) who study the probabilistic context-free grammar given in Figure 4A, consider the branches in a hierarchically generated tree - because two adjacent branches are generated from the same source, knowing a leaf (e.g. /p/) from one branch provides information about the following branch, and therefore about the distribution of leaves in the following branch. If we were to shuffle a branch, the distribution of leaves in that branch would remain the same, but the within-branch organization would be modified. Thus the long-range relationships (between branches) would persist, even when we modify the within-branch structure (which is what our model is doing in Fig 4D, and what we hypothesize is occurring when we shuffle the within utterance phonological organization in Fig 4I).

1.3. Methodology: there is something paradoxical in the methods. The authors claim that children at 6-12 months are still in the babbling stage of development, so have no linguistic structure. But in Figure 3, the authors show that they shuffled words within the transcript, utterances within the transcript, and phones within the utterance -- even for children 6-12 months. But this

presupposes that it is possible to identify words and utterances for children 6-12 months old. In other words, the authors identify three levels of hierarchy while at the same time asserting that hierarchy does not yet exist for this age group.

As we also discuss in 1.1 and 1.2 in the current revision of the manuscript we disambiguate linguistic hierarchy, general hierarchy, and descriptive hierarchy. The datasets we used for this analysis are formatted using the CHAT transcription format (MacWhinney, 2000), which contains a set of rules for determining an utterance under ambiguity (see MacWhinney, 2021, page 59-60). In our analysis, we remain agnostic to whether early speech is generated hierarchically (although we believe most behavior, linguistic or not, is hierarchically generated) or is simply described hierarchically by the researchers using the CHAT format.

We added lines 76-88 to make it clear that we are not claiming that children's speech cannot be described hierarchically, but instead that much of the more complex linguistic hierarchical structure in speech at this age is not yet present.

- 1.4. It could be useful to compare to other statistical methods that have been argued to provide evidence for syntactic combination. See, e.g. Yang (2013) for another way of characterizing the statistical profile of grammatical rules. Applying these methods to the sets of data analyzed here could provide stronger evidence for the (reasonable) assumption in the present paper that young children have not yet acquired complex linguistic structure. Ideally, what you would expect to find would be that the Yang measure becomes more adult-like over time, but that MI is adult-like from the beginning.

We thank the reviewer for their suggestion. We agree it would greatly improve our paper to be able to demonstrate that young children have not yet acquired complex linguistic structure in their productions. Indeed, on the same CHILDES datasets as we used, Meylan et al., (2017) have performed a similar analysis to the Yang (2013) analysis that the reviewer requests. They find that productivity levels are lower at younger ages, suggesting that “[...] children lack rich grammatical knowledge at the outset of language learning [...]”. We added a discussion starting at line 81 which, we agree with the reviewer, supports/provides stronger evidence for the argument that young children have not yet acquired complex linguistic structure.

This paper should ideally be reviewed by someone who can check the statistical methods in more detail. I have just a couple of comments:

- 1.5. Formula (1): shouldn't there be hats on the S's? S without a hat is never defined.

By convention, the hat is meant to distinguish between an estimated and true value, in this case, entropy, S . We follow the notation for Lin and Tegmark, (2017). We added a hat over the S's and I to be more clear (since these are estimates of the true Shannon Entropy and Mutual Information, given finite data).

- 1.5.1. Formula (2): for the distribution [a, b, c], I suppose that $N=3$ and $K=3$. N_i is not defined.

We added this to the text. N_i is the number of elements of category i in the distribution. For example, a naive estimate of entropy can be equivalently written as

$$S(X) = - \sum_{i=1}^n P(x_i) \log P(x_i) \text{ or } S = - \sum_{i=1}^K \frac{N_i}{N} \log \frac{N_i}{N}$$

- 1.5.2. It could be useful to provide simple examples of high and low entropy to give an intuition for how MI models long-distance dependency
 We added a section on hierarchical compression and transition entropy, where we explicitly discuss high versus low entropy relationships (Section 2.2), which leads up to the mutual information section (Section 2.3).
- 1.5.3. I did not follow why "permuting X and Y [a]ffects the joint entropy ... but not the marginal entropies." If a simple example can be given briefly, this might be helpful.
 We added a sentence explaining this to the text. We note, "*The joint entropy is computed over pairs of X and Y, whereas the marginal entropy is computed over X independently and Y independently. Permuting X and Y shuffles the pairing between elements in X and Y, but does not change the frequency of elements in the distributions X or Y.*"
- 1.5.4. Based on the shuffling manipulations, the paper makes the following generalization: "Across each shuffle analysis, we observe that short-range information content captured by exponential decay is largely captured within words and utterances, while long-range information is carried between utterances, even during early language production." While the graphs look somewhat different, I was unsure what the statistical basis for this conclusion was, given that the best model in many cases was still the combined pow_exp model.
 This section of the paper was removed in the current revision to address 2.2. Each of these specific claims is explained in greater detail alongside each shuffling analysis in the paragraph on lines (232-261).

References

- Bolhuis, Johan J., Ian Tattersall, Noam Chomsky, and Robert C. Berwick. 2014. How could language have evolved? PLoS Biology 12.e1001934.
- Pullum, G. K., & Zwicky, A. M. (1988). The syntax-phonology interface. *Linguistics: The Cambridge Survey: Volume 1, Linguistic Theory: Foundations*, 1, 255.
- Yang, Charles. (2013). Ontology and phylogeny of language. PNAS, 110(16): 6324-6327.

2. Reviewer 2

General comments: Overall, this manuscript is clear, well-written, and interesting to read. The authors build off of their previous work in adult human speech and animal communication, and contextualize their research in the broader topic of sequential behaviors. The authors provide compelling evidence for long-range dependencies and hierarchical structure in the vocalizations of developing children that are reminiscent of those observed in adult speech.

Major comments:

2.1. While I think their findings are very interesting, I'd like the authors to speak more about the novelty of the work. Their results suggest that adult-like hierarchical structure can be observed even in the simple grammars of children: the utterances of children have some repeatable and predictable sequence structure, therefore can be described by Markovian and/or power-law decay functions. The authors are correct to highlight that hierarchical structure is also observed in non-linguistic human behaviors as well as in the vocalizations of non-human animals. So, how novel/unexpected is it that the speech patterns of children, even infants and toddlers, contain hierarchical structure?

We added sections to both the introduction (lines 65-88) and discussion (lines 293-300) to make the novelty of these results more explicit. In particular, we discuss why under previous hypotheses these long-range relationships might not be expected. Early vocalizations are not underlied by the same structured syntax or discourse as adult speech, which are hypothesized to underlie long-range correlations. At the same time, as the reviewer mentions, there is reason to expect our results: long-range relationships are fairly common phenomena independent of language, and have been observed in a number of non-linguistic behaviors and signals.

2.2. The authors should discuss how the parameters of the power-law, exponential decay, and composite models compare between the current dataset and the corpus of adult speech from Sainburg et al. (2019), as well as age-dependent variation in the distance (in phones) at which the MI decay transitions from exponential to power law. I can appreciate the complexity of such comparisons across different levels of speech, but these comparisons would allow readers to assess how "adult-like" these long-range statistical relationships are in developing children.

We agree with the reviewer that an analysis of how these parameters change with age would help in understanding how adult-like the observed long-range relationships are. As we discuss in 2.3, we perform this analysis.

We find that much of the variation in parameters are the result of exogenous factors, most notably that datasets at different ages were largely acquired from different research groups and settings, meaning datasets and dataset sizes can vary substantially. That variance in dataset size strongly impacts the relative contribution of power-law and exponential parameters, and makes it difficult to distinguish how these parameters are changing with age. Below is one example of the relative proportion of the decay model explained by the exponential component (left) and the power-law component (right). As is shown in the next section, this shift toward an increase in the exponential component explaining the model at short distances, and the power-law explaining the model at long distances with age, is confounded by the availability of increasingly larger datasets (longer transcripts) with age. A more complete view is given in 2.3.

Fig L1: Proportion of model explained by exponential component (left) and power-law component (right) of fit decay model for PhonBank for the English datasets.

2.3. The authors write that “long-range dependencies in language and other human behavior [33, 60, 61] may reflect more general biological processes inherited from the organization of underlying neurophysiological mechanisms [62–65].” In order to distinguish the contribution of linguistic input (learning) from fundamental (“innate”) biological contributions, it would be revealing to compare the speech of at least one group of children that speak a different language. For example, because the shape of the MI functions seem to differ between adult English, German, and Japanese-speaking adults (Sainburg et al., 2019), comparing the English-speaking children to Japanese- or German-speaking children would be very informative. If the shape of the curve in children is a function of biology and not linguistic exposure, then the shapes should be statistically indistinguishable across children that speak different languages with different MI functions.

We performed the analyses requested.. As intuited by the reviewer, the results do yield some interesting insight into the relationship between development, linguistic exposure, and MI decay. We think this direction of analysis is very promising, however we do not believe that with our current dataset we can adequately interpret our data in this light, due to limitations in the variability of dataset size across age groups. Our primary issue is variability in corpus lengths between age groups. Below, we first show an analysis of decay model parameters without controlling for corpus length and the effect of corpus length on those decay models. We follow that analysis with an analysis where we control for corpus length by sub-sampling datasets for each age group. We find an interesting effect, where we observe that the exponential decay parameter changes with age, but the power-law parameter remains constant. We included Figure L3 in the supplementary materials as well. We also added the MI decay analysis in French as a supplementary figure, which fully replicated our results in English.

In the figure below, we show each parameter in the composite decay model:

$$a * e^{-x*b} + c * x^d + f$$

as a function of age (top two rows) and as a function of dataset size (bottom row). In the top row, the lines correspond to the model fits within age groups, the dots correspond to the model fits for the longest individual transcripts in the PhonBank corpora, and color corresponds to the language in the top row.

Fig L2: Fit decay model parameters across age groups and for individual transcripts. (top) Scatterplot of the decay-fit parameters of the single longest transcripts in both the French (blue) and English (red) datasets. The line plots indicate the same parameter fits, across the entire datasets. (middle) A linear model fit to the same data, for the parameter value as a function of age. (bottom) A linear model fit to the parameter values as a function of the number of phonemes in the dataset.

To attempt to tease apart the effect of corpus length and age group on parameters, we sub-sampled corpora at varying lengths up to the total length of the corpus for each of the 100 largest corpora for each age group, and plotted the resulting parameter fits by subsampled corpus size for each age group. Some effect of age group on the exponential decay parameter is observed, but there is little to no effect on power-law decay parameter.

Fig L3: Fit decay parameters over each age group for the exponential decay parameter, power-law decay parameter, and exponential to power-law transition. Lines plotted are median across the 100 largest transcripts for each age group. Median is used to discount poor model fits for individual transcripts. Color corresponds to age group, where purple is 6-12 months and red is 3+ years.

While we observe some effect of age on the parameters, we are not fully confident that we have a sufficient dataset to differentiate these trajectories from the effects of the limited data that we have available. Future work with larger corpora (e.g. the Speechome corpus) would likely give better estimates of the trajectory of these parameters.

We added a discussion of how such an analysis could extend our understanding of the origins of these long-range relationships in a future-directions section (lines 301-312).

Minor comments:

- 2.4. In the last sentence of the abstract, the authors write that “(t)hese linguistic structures cannot, therefore, be the sole cause of long-range statistical dependencies in language.” It is not really clear about the degree to which the authors are claiming that these long-range dependencies in infant vocalizations are innate. One question is to what degree long-range dependencies can be inferred from auditory patterns heard in utero. There is increasing evidence that auditory experiences in utero shape the brain and vocal output in humans, and it would be useful for the

authors to discuss the extent to which in utero experiences could allow infants and toddlers to communicate with hierarchical structure.

We agree with the reviewer that it is possible that the long-range relationships in infant vocalizations are shaped by in-utero experiences. We now mention this in the discussion: *“It is also possible that the observed long-timescale structure present in early childhood vocalizations are in some way influenced by linguistic exposure that precedes vocal production, such as in-utero exposure to speech”*.

2.5. Line 182-185: The authors should provide information about how prevalent repeated elements are at different levels (e.g., words, utterances) and different ages. This would provide some insight into the importance of this set of analyses

We added this information to the caption for the corresponding figure.

Phone, word, and phrase repeats all decrease as with age (words: 1-1.5 years=14.3%, 1.5-2=7.4%, 2-2.5=3.7%, 2.5-3=3.2%, 3+=1.9%; phrases: 1-1.5 years=18.0%, 1.5-2=13.2%, 2-2.5=7.2%, 2.5-3=5.3%, 3+ =1.8%; words: 0.5-1 years=2.5%, 1-1.5=2.9%, 1.5-2=2.1%, 2-2.5=1.0%, 2.5-3=0.8%, 3+ =0.8%). Thus the importance of removing repeats is greater in transcripts from younger children. In addition, in younger age groups many of the phones do not correspond to words at all. In the PhonBank datasets, for children in the 6-12 months age group 98.7% of phones were coded as either corresponding to unintelligible words, or phonologically coded without reference words (other age groups: 1-1.5=68.0%, 1.5-2=42.6%, 2-2.5=20.9%, 2.5-3=16.1%, 3+ =12.3%).

Appendix B

Dear Dr. Barton,

Thank you for accepting our manuscript for publication. Below, we have responded to each of the reviewer's comments and requests, which are reflected in this revision of our manuscript.

Thank you,

Tim Sainburg

Anna Mai

Tim Gentner

0. Comments to Author:

Thank you for the careful revision of the manuscript. Both the reviewers and I find the manuscript greatly improved. However, both reviewers still see room for additional improvement and clarification.

- 0.1. Reviewer 1 suggests the French analysis would be more useful if it could be shown that French has a different MI decay structure than English. If this information is not readily available, it would be good to discuss this limitation. Reviewer 1 also suggests ways to directly compare the organizational structure of adult and children's speech.

The reviewer suggests the addition of an interesting and important line of questioning that extends upon our current results. The current study demonstrates the presence of long and short-range statistical dependencies in speech originating early in development. The reviewer suggests that by (1) looking at multiple languages and (2) looking at how MI decay parameters change throughout language development, we can develop an understanding of whether and how statistical relationships in speech change with exposure to language.

We agree that this is a potentially interesting set of questions, now rendered important by our initial observations of early developmental vocal structure. Limitations in the current datasets place these additional analyses beyond the scope of the current work, however, and we respectfully suggest that the analyses are better left for future work.

- 0.2. Both reviewers suggest improving or (or possibly removing) Figure 2 A-C or, at least, improving the description of it.

We moved Figure 2 A-C to the supplemental materials and improved its description in the caption.

- 0.3. Both reviewers make several other suggestions for clarification.

The remaining suggestions involved the rewording of sentences or changes in figure structure. We have added clarifications and edited sentences where reviewers have requested.

1. Referee: 2

Comments to the Author(s).

GENERAL COMMENTS: The revised manuscript clarifies a number of important aspects outlined by the reviewers. I appreciate the additional analyses and points in the Discussion section. In my opinion, the manuscript remains an excellent contribution to our understanding of the sequential and hierarchical organization of behavior. I have some outstanding issues that should be clarified before the article is published.

1.1. I like the addition of French in the analysis as an additional data point for children but, without knowing the hierarchical structure of adult French, it is not clear if this analysis satisfied my concern. I understand that their analyses are limited to the availability of annotated corpora, but I was hoping that the authors could analyze the developmental pattern of hierarchical structures for languages that have MI decay functions that are distinct from adult English (e.g., adult German based on figures in Sainburg et al., 2019). It would be interesting to see if children raised with languages that have different MI decay functions (as determined by the analysis of adult speech) also demonstrate different MI decay functions (or are organizational differences only found in adult speech). Have the authors analyzed the sequential organization of adult French in this manner? If so, does the hierarchical structure of adult French differ from adult English?

In 1.1 and 1.2 the reviewer brings up two interesting and related lines of questioning: by looking at additional languages with different MI decay structure (e.g. German as observed in Sainburg et al., 2019) it would be possible to quantify the changes in MI decay structure over development, from decay parameters presumably shared across young children with limited language experience, to adulthood, where differences in MI decay parameters would emerge. We agree that this would be a very interesting line of questions, but for the reasons noted below we respectfully suggest that it is beyond the scope of the present work. We note a limitation of the current dataset in addressing this question is in response 1.2.

We agree that the line of questioning raised by the reviewer is very interesting and important to ask. It is unclear, however, whether the data required to address it exist. To properly ask this question, ideally, the datasets used would provide MI decay fits across several different language backgrounds each expressing different decay parameters in adult speech. Importantly, the transcripts would need to control for differences in recording settings (i.e. interactions and prompting from interviewers and experiments would need to be held constant or be uniformly randomized). In addition, transcripts would need to be sufficiently long, and of similar lengths across all age groups (see response 1.2). Thus the adult datasets used in Sainburg et al., (2019) alongside the developmental datasets (TalkBank) used in the current work could in principle be used to make inferences about changes in MI decay structure, but limitations include dataset sizes for younger children (as is discussed in 1.2), and differences in recording settings across transcripts (transcripts for each language are generally recorded in varying settings).

1.2. In the previous round, I recommended that the authors discuss how “adult-like” the organizational patterns of children speech was by plotting the data for adults in the same figure as for children (either in the main text or supplementary information section). A direct comparison of this nature is important to understand how the temporal patterning of communication changes over time, to think about mechanisms of behavioral control and organization, and to understand what the authors exactly mean by “adult-like”. In addressing my suggestion, the authors discuss the complexities of such an analysis given differences in transcript sizes across corpora and the effects of transcript size on parameter estimates (e.g., Figure S3). However, isn’t it possible to analyze adult corpora that have been filtered or pruned in such a manner to be equivalent in transcript sizes as those for children? Or to run an analysis between adult and children where transcript size is used as a covariate?

The reviewer asks about the possibility of additional analyses of how adult-like children’s speech is, based on a comparison of MI decay model fit parameters between adult speech and child speech. They suggest controlling for transcript length either by making adult transcript lengths shorter for direct comparison, or adding transcript size as a covariate in e.g. a linear model predicting decay parameters. Given the effects we have shown, we agree that a quantitative assessment of the degree of similarity between child and adult speech is potentially valuable. For reasons outlined below, however, limitations of the current dataset make the reliability of such an analysis problematic. Thus we think it is most prudent to target this question in future work with access to more well-controlled datasets.

In the previous revision, we tried to get at these questions by adding two analyses that varied the length of transcripts to enable direct comparisons between younger age groups and older age groups. The oldest age group (3-4 years) are not adults, of course, but longer transcripts are available for them. Results of the first analysis, which looks at hierarchical compressibility while varying transcript size, are shown in Fig 2E-F. Results of the second analysis, which (as the reviewer notes) considers decay fit parameters as transcript size varies, are shown in Fig S4. Although these are not precisely what the reviewer is now asking for, the overall purpose of these analyses is to estimate a trajectory of decay fit parameters over development, as the reviewer is requesting. As the results in Figure S4 show, however, the shorter transcripts available for younger children yield a high degree of uncertainty in parameter estimates. Likewise, reducing transcripts lengths in older age groups to match the younger age groups yields similarly high degrees of uncertainty in parameter estimates. This pattern suggests that we are coming up against a hard limitation of the current data due to varying transcript sizes. While, in principle, it would be possible to fit a linear model to the data, using transcript length as a covariate, we would not be comfortable trying to interpret the significance of any quantitative statistical differences (or lack of differences) that might result. Instead, we contend that these types of follow-up questions are best addressed by analyses over different datasets sampling longer transcripts of children’s speech where transcript size is more

well-controlled. To help avoid any confusion surrounding these points in the revised manuscript, we clarified our use of the phrase “adult-like” (discussion, first paragraph, see 1.7), to refer to the presence of long and short-range structure, rather than language-specific decay parameters.

MINOR POINTS:

- 1.3. Figure 2A-C. The authors should provide more information on this plot. How should one interpret the position of circles and distance of circles from each other (if the dimensions and distances meaningful at all). I assume the thickness/darkness of the arrows indicates transition probability but that is not mentioned in the figure legend. While this graphical depiction is more interesting than a table, readers might be more used to seeing a table with transition probabilities between phones to summarize first-order transitions.

We moved the graph panels to the supplementary material. In addition, we added text explaining the layout of the graph to the figure caption.

- 1.4. Line 248-251: I realize that these sentences refer to different panels in Figure 4 but the sentences overlap so much that it's hard to differentiate. Maybe combine into one sentence? Or maybe re-write in a manner that allows readers to better differentiate.

Following the reviewer's suggestion, we broke the paragraph up into three separate paragraphs to clearly disambiguate the three relationships between model and data panels (C to G, D to H, E to I).

- 1.5. Line 251-253: indicate what is the parallel for this type of shuffling? (e.g., like the authors did in lines 247-248: Shuffling Markov sequences within the transcript (Figure 4C) relates to shuffling words within the transcript (4G).”)

The referenced text is:

On the opposite end of the hierarchy, when Markovian structure is destroyed by shuffling the sequence within the Markov-generated sequences (Figure 4D), the exponential component of the decay is lost (dashed line) and the decay is dominated by the power law.

As is mentioned above, the paragraph structure now indicates more clearly 4D is related to 4H.

- 1.6. Figure 4 (and Figures S4, S5). It is sometimes hard to reconcile the main text with indications of best model fits above the panels. For example, in lines 254-255, it is mentioned that shuffling phones within utterances reduces the exponential component; consequently, one might intuit that the best function would be simply the power law function. But in Figure 4I, the power-exponential model provides the best fit for each age

group. (I understand that the exponential component is reduced, not necessarily eliminated.) So, I suggest the authors add additional information to the panels (instead of just listing the best model), maybe the AIC values for each of the different models. (I acknowledge that this information is in Tables 1 & 2.). Basically, I would like the information in the figure to match the text better.

We added the AIC values to the figure panels to provide additional information beyond best fit model. In addition, we broke the paragraph referring to figure 4 up into individual paragraphs to make it easier to connect the text, model panels, and data panels.

- 1.7. Line 285: not clear what “adult-like” means. Adding a few sentences describing the structure of adult speech at the beginning of the Discussion would be useful.

The referenced paragraph is:

*We analyzed the developmental time course of long-range sequential information present in speech throughout early childhood. We observed **adult-like** long-range statistical relationships \cite{sainburg2019parallels} present as early as 6 to 12 months in phoneme sequences, and at 12-18 months in word sequences. Our findings compel reconsideration of the mechanisms that shape long-range statistical relationships in human language. Traditionally, the power-law decay in information between the elements of language (phonemes, words, etc.) has been thought to be imposed by the hierarchical linguistic structure of syntax, semantics, and discourse \cite{alvarez2006hierarchical, lin2017critical, altmann2012origin}. Early speech development provides a natural experiment in which one can examine human vocal communication absent the production of complex syntactic and discourse structure \cite{greenfield1991language, meylan2017emergence, crain2012syntax, ravid2005emergence, berman2002cross}. Remarkably, even at a very early age, prior to the production of mature linguistic structures, vocal sequences show adult-like long-range dependencies.*

Following the reviewer’s suggestion, we changed the text (in bold) to:

*We analyzed the developmental time course of long-range sequential information present in speech throughout early childhood. We observed adult-like long-range statistical relationships \cite{sainburg2019parallels} present as early as 6 to 12 months in phoneme sequences, and at 12-18 months in word sequences. Our findings compel reconsideration of the mechanisms that shape long-range statistical relationships in human language. **Adult speech is characterized by both long-range and short-range statistical relationships characterized by power-law and exponential decays in mutual information, respectively.** Traditionally, the power-law decay in information between the elements of language (phonemes, words, etc.) has been thought to be imposed by the hierarchical linguistic structure of syntax, semantics, and discourse \cite{alvarez2006hierarchical, lin2017critical, altmann2012origin}. Early speech development provides a natural experiment in which one can examine human vocal communication absent the production of complex syntactic and discourse structure \cite{greenfield1991language, meylan2017emergence, crain2012syntax, ravid2005emergence, berman2002cross}. Remarkably, even at a very early age, prior to*

the production of mature linguistic structures, vocal sequences show adult-like long-range dependencies.

- 1.8. Line 306-307: the wording here should be more consistent with the analysis. For example, the authors could write: “Although including information about the extent of adult speech with children in the corpora did not increase explanatory power, ...”. The current wording is too vague and subject to over-interpretation.

The referenced text was:

While we did not observe that long-range relationships were driven by interactions with adult speakers, we cannot completely rule out language interaction or other exogenous variables as possible drivers for the observed long-range relationships.

We changed the text to

*While our results **did not provide evidence** that long-range relationships are driven by interactions with adult speakers, we **cannot rule out** language interaction or other exogenous variables as possible drivers for the observed long-range relationships.*

- 1.9. Figure S2. Add age groups to the tops of each panel. Better yet, why not replot the figures for English above or below the plots for French.

As requested, we added the age groups and the English panels above Figure S2 (now S3).

- 1.10. Figure S4 vs. S5. When you shuffle between utterances in Figure S4, the power-exponential model provides the best fit for all groups. However, when you shuffle between utterances in Figure S5, there is more variability in the best fit. Please discuss.

As noted above, we added the AICc values to aid in understanding differences in model fit, rather than displaying the best fit model alone. This makes it easier to understand the variability noted by the reviewer. In both of the examples the reviewer mentions, although there is variability in the best fit model, the AICc shows that in each panel, the combined model provides little extra explanatory power than the exponential model alone (in contrast with e.g. Fig S6A where the AICc difference is much greater in support of the combined model).

- 1.11. Figure S7 and S8 are not cited in the main text, though maybe S7 and S8 should be references on line 265?

All supplementary figures are now cited in the text.

2. Referee: 1

Comments to the Author(s).

Summary:

Human language is known for displaying hierarchical structure. The relation between linguistic hierarchy and other hierarchical organization in human and non-human behavior is a rich area of recent research. The present paper investigates one recent statistical method that has been used to

detect and compare hierarchical structure for human and non-human animals (a power-law decay of long-distance dependencies). The results of the paper provide important new insights into how to best interpret these statistical findings (namely, they must be explained at least in part by organizational mechanisms outside of linguistic hierarchy).

Evaluation:

I reviewed a previous version of this paper. Overall, I find the new version of the paper to be greatly improved. The newly added paragraphs in the discussion section adequately address my previous concerns regarding the lack of a positive proposal; the discussion section now provides several different explanations of the data that could be investigated in further work. The new subsection in the introduction on different kinds of hierarchy is useful for framing the goals of the paper. The comparison of the shuffle procedure to the same procedure on an idealized model is useful for understanding the effects found.

I have a few remaining concerns:

- 2.1. First, I found it rather difficult to understand the goals of the newly added sections 2.2 and 3.1. The conclusion from section 3.1 is that "across all age groups, transcripts are more hierarchically compressible than their Markov or randomized counterparts" and that this is "similar across each age group." Is this then another way (beyond the MI analysis) of showing that hierarchical organization exists before the acquisition of linguistic structure? If so, then what does the MI analysis show beyond what we'd expect from the results using Sequitur hierarchical compressability? The paper would be improved if the authors could more clearly lay out how these two analyses relate to each other, and what insights each gives about the hierarchical structure of the data.

Figure 2, added in the previous revision, was added as a scaffold in understanding the conceptually more challenging information-theoretic analysis: first, we show that there is non-random structure in the signal, then we show that the signal can be more compressively hierarchically described than a Markov model. Finally, in the information-theoretic analysis, we characterize the short and long-range organization in the signal.

To more clearly lay out how these two analyses relate to each other we added a sentence in the mutual information results section:

These results indicate non-random and hierarchical structure across corpora from all age groups. In the following mutual information analyses, we will measure the sequential relationships underlying these signals.

- 2.2. Second, in section 3.3, comparison to an ideal model provides a very useful visual baseline for understanding what is going on in the shuffle analyses, but I think it bears noting that this section of the paper does not report any statistical tests. I'm not sure how problematic this is for the paper, given that there are statistical tests reported for the main

results in section 3.2 (Tables S1-S2). But this feels like a somewhat important gap. (Note also that statistical results are not presented in section 3.1 but probably could be.)

We added the AIC values for each fit model to the panel figures, which enables comparison on the basis of relative probability, as in the main analyses. A $\Delta\text{AIC} \geq 2$ is sometimes used as a threshold in a similar manner as $p=0.05$ in model selection, though we choose not to do so here following the recommendation by Burnham et al., (2011; cited in manuscript).

- 2.3. Finally, the idealized model in Section 3.3 suggests that phonological relations are Markovian but that syntactic relations are hierarchical. If this is true, then it is perhaps somewhat surprising that there is still a significant exponential component to the MI functions in the analysis based on word distance (Figures 3DEF). This is potentially related to the finding that, when phones are shuffled, the best model still contains an exponential component (Figure 4I). Some discussion of this point could potentially be clarifying.

This is an interesting point and is consistent with our previous paper (Sainburg et al., 2019). One explanation is that phonological relationships extend across words. To make it clear that we are not proposing that either words or utterances are the boundaries between Markovian and hierarchical organization, we added this sentence to the Figure 4 results section:

Consistent with the analyses over words in the CHILDES transcripts (Figure 3F), the exponential component of the decay is not entirely destroyed when shuffling within utterances (Figure 4H) or words (Figure S5C), indicating that a boundary between Markovian and hierarchical organization cannot be fully assigned at either level of organization.

- 2.4. Minor comments:

- 2.5. Line 197: is "interred" the intended verb here?

We fixed the typo (inferred).

- 2.6. Figure 2: I'm not sure how useful the example Markov models are. They may be too messy to provide any meaningful information.

We moved these panels of the figure to the supplementary materials.

- 2.7. Figure 3: It is currently confusing why the 6-12 month age group is excluded for the MI analyses for word distance, given that words are coded for this age group (as seen in the shuffle analysis). I think ultimately the answer to this is rather mundane: namely, the analysis of word distance used CHILDES data instead of Phonebank data, and CHILDES data starts at 12 months. Perhaps this confusion could be avoided by mentioning this fact explicitly in the caption of the Figure. (Lines 285-286 similarly implicate that the paper

failed to find long-range statistical relationships in word sequences. It just wasn't tested.)

A footnote in the methods section reads “No 6-12 month age group was used in word-level analyses because of the sparsity of word-level productions at that age“. We repeated this sentence in the caption for clarity.

2.8. Line 225: Table S3, I think, not Table 3.

We fixed this error.

Figure 4:

2.9. -Please move the age-group labels ("6-12 months", etc.) to a more noticeable place, possibly below the last graphs.

We moved the labels below the last graph.

2.10. -The labels I and H should be swapped to respect alphabetical order.

We swapped I and H.

2.11. -The labels used in the caption currently do not correspond to the labels of the figures.

We fixed this error.

2.12. -It should be clarified that when the exponential model is the best fit, the exponential component (the only component) is a solid line and the power-law component is not shown.

We added this to the caption.

2.13. Line 232-233: "the PCFG model from Figure 1". Do the paper or supplemental materials ever provide the precise PCFG that was used to generate the graphs in Figure 1 Figure 4ABCDE?

We added the parameters for the PCFG model from Figure 1 to the supplementary materials. Standalone Jupyter notebooks to fully reproduce these figures are also available in the code.